# Cyclin-dependent kinase 5 (CDK5) regulates the circadian clock

Andrea Brenna, Iwona Olejniczak, Rohit Chavan, Jürgen A Ripperger, Sonja Langmesser, Elisabetta Cameroni[†], Zehan Hu, Claudio De Virgilio, Jörn Dengjel, Urs Albrecht*

Department of Biology, University of Fribourg, Fribourg, Switzerland

**Abstract** Circadian oscillations emerge from transcriptional and post-translational feedback loops. An important step in generating rhythmicity is the translocation of clock components into the nucleus, which is regulated in many cases by kinases. In mammals, the kinase promoting the nuclear import of the key clock component Period 2 (PER2) is unknown. Here, we show that the cyclin-dependent kinase 5 (CDK5) regulates the mammalian circadian clock involving phosphorylation of PER2. Knock-down of *Cdk5* in the suprachiasmatic nuclei (SCN), the main coordinator site of the mammalian circadian system, shortened the free-running period in mice. CDK5 phosphorylated PER2 at serine residue 394 (S394) in a diurnal fashion. This phosphorylation facilitated interaction with Cryptochrome 1 (CRY1) and nuclear entry of the PER2-CRY1 complex. Taken together, we found that CDK5 drives nuclear entry of PER2, which is critical for establishing an adequate circadian period of the molecular circadian cycle. Of note is that CDK5 may not exclusively phosphorylate PER2, but in addition may regulate other proteins that are involved in the clock mechanism. Taken together, it appears that CDK5 is critically involved in the regulation of the circadian clock and may represent a link to various diseases affected by a derailed circadian clock.

*For correspondence:
urs.albrecht@unifr.ch

Present address: †Humabs Biomed SA, Vir Biotechnology, Bellinzona, Switzerland

## Introduction

The circadian clock, prevalent in most organisms, is an evolutionary adaptation to the daily light-dark cycle generated by the sun and the earth's rotation around its own axis (*Rosbash, 2009*). This clock allows organisms to organize physiology and behavior over the 24 hr time scale in order to adapt and thus optimize, body function to predictably recurring daily events. Malfunctioning or disruption of the circadian clock in humans results in various pathologies including obesity, cancer, and neurological disorders (*Roenneberg and Merrow, 2016*). In order to maintain phase synchronicity with the environmental light-dark cycle, the suprachiasmatic nuclei (SCN), a bipartite brain structure located in the ventral part of the hypothalamus above the optic chiasm, receive light information from the retina. The SCN convert this information into humoral and neuronal signals to set the phase of all circadian oscillators in the body (*Dibner et al., 2010*).

In order to measure the length of one day, organisms have developed cell-based molecular mechanisms relying on feedback loops involving a set of clock genes. The existence of such loops was suggested by the analysis of *Drosophila* having various mutations in their *period* (*per*) gene (*Hardin et al., 1990*). Further studies completed the picture of intertwined transcriptional feedback loops at the heart of the *Drosophila* circadian oscillator (*Darlington et al., 1998*). Every day, per accumulates to a certain concentration upon which it enters into the nucleus together with timeless (tim). This protein complex inhibits transcriptional activation mediated by dClock and cycle acting on the expression of *per* and *tim*. After the degradation of the inhibitor complex, the repression is relieved and a new circadian cycle starts.

**eLife digest** Anyone who has crossed multiple time zones on a long flight will be familiar with jet lag, and that feeling of wanting to sleep at lunchtime and eat in the middle of the night. Many bodily processes, including appetite and wakefulness, roughly follow a 24-hour cycle. These cycles are known as circadian rhythms, from the Latin 'circa diem' meaning about a day. An area of the brain called the suprachiasmatic nucleus (SCN) coordinates circadian rhythms. It acts as a master clock by generating a 24-hour signal for the rest of the body to follow. Jet lag occurs when this internal circadian rhythm becomes out of sync with the local day-night cycle.

Although jet lag can be uncomfortable, it tends to disappear over the course of a few days. This is because exposure to daylight in our new location resets the SCN master clock, enabling us to adapt to a new time zone. But evidence suggests that long-term disruption of circadian rhythms, for example as a result of shift work, may have lasting harmful effects. These include an increased risk of degenerative brain disorders such as Parkinson's disease and Alzheimer's disease.

Brenna et al. now identify a molecular mechanism that could explain this link. A key component of the SCN master clock is a protein called Period2 (PER2). Levels of PER2 rise and fall over each 24-hour period, helping the brain keep track of time. Brenna et al. show that PER2 interacts with CDK5, a protein that helps regulate brain development and that has been implicated in Parkinson's disease and Alzheimer's disease. Reducing CDK5 levels in mice shortened their circadian rhythms by several hours. It also altered the animals' behavioral patterns over a 24-hour period. Deleting the gene for PER2 had a similar effect, suggesting that CDK5 helps regulate PER2.

Future studies should investigate the molecular links between CDK5, circadian rhythms and processes such as neurodegeneration. The results would provide clues to whether manipulating the circadian clock could help prevent or treat neurological disorders.

To fine-tune the period of the circadian oscillator, kinases regulate the accumulation and nuclear entry of per and tim. The kinase double-time (dbt) phosphorylates per to destabilize it and to prevent its transport into the nucleus (*Kloss et al., 1998*; *Price et al., 1998*). On the other hand, the kinase shaggy (shg) phosphorylates tim to stabilize the heterodimer and to promote its nuclear translocation (*Martinek et al., 2001*). Many other kinases and phosphatases are necessary to complete the *Drosophila* circadian cycle and to adjust its phase to the external light-dark rhythm (*Garbe et al., 2013*).

The circadian oscillator of mammals is arranged very similarly to the one of *Drosophila*, with some modifications (*Dibner et al., 2010*; *Takahashi, 2017*). For instance, the function of *Drosophila* tim to escort per into the nucleus was replaced by the Cryptochromes (Cry) in the mammalian system (*van der Horst et al., 1999*). Furthermore, the first mutation to affect the mammalian circadian oscillator, *Tau*, was later mapped to Casein kinase Iε (CK1ε), which is the *Drosophila* dbt orthologue (*Lowrey et al., 2000*). One of the sites phosphorylated by CK1ε within human PER2 is mutated in the Familial Advanced Sleep Phase Syndrome (FASPS) (*Toh et al., 2001*). This mutation and also the *Tau* mutation were subsequently introduced into the mouse genome to prove their functional relevance (*Meng et al., 2008*; *Xu et al., 2007*). However, a kinase similar to the function of shg in *Drosophila*, which stabilizes and promotes the import of PER proteins into the nucleus of mammals (*Hirano et al., 2017*), has not been identified. Interestingly, PER2 contains over 20 potential phosphorylation sites (*Vanselow et al., 2006*), indicating that mammalian PER and specifically PER2 are highly regulated at the post-translational level. This degree of phosphorylation is probably contributing to the precise rhythmicity of PER2, which stands out as a crucial feature of the core clock (*Chong et al., 2012*).

Among the plethora of kinases identified that phosphorylate mammalian clock proteins, cyclin-dependent kinase 5 (CDK5) was found to target CLOCK (*Kwak et al., 2013*). CDK5 is a proline-directed serine-threonine kinase belonging to the Cdc2/Cdk1 family that is controlled by the neural specific activators p35, p39 (*Tang et al., 1995*; *Tsai et al., 1994*), and cyclin I (*Brinkkoetter et al., 2009*). CDK5 regulates various neuronal processes such as neurogenesis, neuronal migration, and axon guidance (*Kawauchi, 2014*). Outside of the nervous system CDK5 regulates vesicular transport, apoptosis, cell adhesion, and migration in many cell types (*Contreras-Vallejos et al., 2012*). It has

been proposed that CDK5 modulates the brain reward system (*Benavides et al., 2007*; *Bibb et al., 2001*) and that it is consequently linked to psychiatric diseases (*Engmann et al., 2011*; *Zhu et al., 2012*). Interestingly, the clock components PER2 and CLOCK have been associated with the same processes (*Abarca et al., 2002*; *Hampp et al., 2008*; *Roybal et al., 2007*). However, it is unknown whether CDK5 plays an important role in the central oscillator of the circadian clock.

In this study, we wanted to identify proteins promoting the nuclear transport of PER2 with focus on kinase(s) acting similarly to shg. Using a genetic synthetic lethal dosage screen in yeast, we observed a genetic interaction between *Per2* and *PHO85*, which encodes a cyclin-dependent protein kinase that is orthologous to CDK5 in mammals. Subsequent experiments in mice demonstrated that silencing of *Cdk5* in the SCN shortened the clock period. Our study identified CDK5 as a critical protein kinase in the regulation of the circadian clock and in particular as an important regulator of the crucial clock component PER2.

## Results

### Genetic interaction between Per2 and CDK5 in yeast and diurnal activity of CDK5

In order to gain insight into the regulation of PER2 function in mice, we initially tried to identify genes that genetically interact with *Per2* in budding yeast by using a variation of the Synthetic Genetic Array (SGA) method (*Tong, 2001*). To this end, we carried out a synthetic dosage lethality (SDL) screen, which is based on the concept that a high dosage of a given protein (i.e. PER2 in this case) may have negligible effect on growth in wild-type cells (as we found to be the case for PER2; *Figure 1A*), but may compromise growth in mutants that have defects in pathway components or in functionally related processes (*Measday et al., 2005*; *Sopko et al., 2006*). Of note, SDL screens have been instrumental in the past to specifically predict the relationship between protein kinases and their targets (*Sharifpoor et al., 2012*). Our search in a yeast knockout collection (encompassing 4857 individual deletion strains) for mutants that exhibited significantly reduced growth when combined with increased dosage of PER2 (see Materials and methods for further details) allowed us to isolate three mutants, namely eap1Δ, gnd1Δ, and pho85Δ (*Figure 1A*). Among these, the strain lacking the cyclin-dependent protein kinase Pho85 was most dramatically compromised for growth in the presence of high doses of PER2. Hence, Pho85 antagonizes the growth-inhibitory effect of PER2 in yeast, which indicates that the Pho85-orthologous CDK5 may potentially act upstream of PER2 in mammalian cells.

The protein kinase CDK5 is mostly expressed in the brain and has previously been implicated in phosphorylation of mammalian CLOCK (*Kwak et al., 2013*). However, the functional relevance of CDK5 for the clock mechanism has never been tested. Therefore, we investigated whether CDK5 affected the functioning of the circadian clock. First, we assessed whether CDK5 displayed time of day-dependent expression and activity in the SCN, the master clock of the circadian system. We collected SCN samples every 4 hr starting from ZT0 until ZT20 (ZT0 = light on, ZT12 = light off), and performed western blots on total extracts using specific antibodies (*Figure 1B*). The immunoblot against CRY1 showed a diurnal profile of this protein with a peak during the late-night phase, confirming that the mice were entrained properly to the light-dark cycle. In contrast, the CDK5 accumulation profile seemed to be unaffected by the time of day (*Figure 1B*). Next, we investigated whether CDK5 kinase activity displayed a diurnal profile. While CDK5 levels did not change significantly over one day (*Figure 1B*), we observed that histone-H1, a known CDK5 target (*Peterson et al., 2010*), was phosphorylated by this kinase in a time of day-dependent manner, with the highest levels of CDK5 activity observed at ZT12 to ZT20, that is during the dark phase (*Figure 1C*). Phosphorylation of histone-H1 was specifically blocked by roscovitine, a CDK5 inhibitor (*Hsu et al., 2013*), whereas LiCl, a Gsk3β inhibitor, did not affect this phosphorylation (*Figure 1D*), suggesting a CDK5-specific phosphorylation. Altogether, these data demonstrated that CDK5 kinase activity (but not protein accumulation) was diurnal in the SCN.

### CDK5 regulates the circadian clock

Since CDK5 activity displayed a diurnal profile in the SCN, we tested whether knock-down of CDK5 in the master clock of the SCN changed circadian behavior in mice. To this end, we tested various

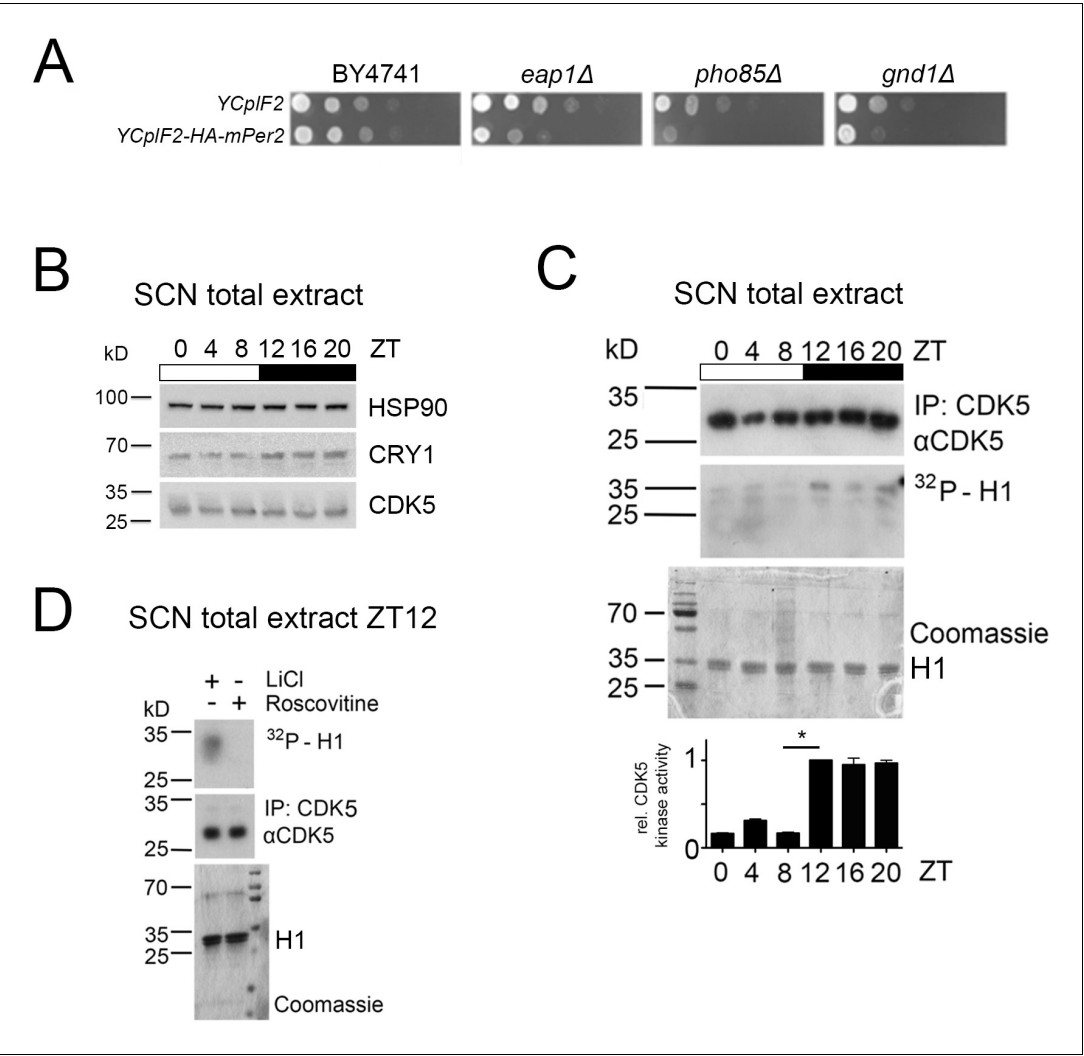

**Figure 1.** CDK5 intersects with PER2 and has diurnal activity in the SCN. (**A**) Loss of Eap1, Gnd1, or Pho85 compromises growth of PER2-overproducing yeast cells. The yeast mutants *eap1Δ*, *gnd1Δ*, and *pho85Δ* were identified in a synthetic dosage lethal screen as detailed under Methods. Wild-type (BY4741) as well as *eap1Δ*, *gnd1Δ*, and *pho85Δ* mutant cells carrying the control plasmid (YCpIF2) or the YPpIF2-*mPer2* plasmid (that drives expression of mouse PER2 from a galactose-inducible promoter) were pre-grown on glucose-containing SD-Leu media (to an $OD_{600}$ of 2.0), spotted (in 10-fold serial dilutions) on raffinose and galactose-containing SD-Raf/Gal-Leu plates, and grown for 3 days at 30°C. (**B**) Immunoblot was performed on SCN extracts around the clock. SCN from seven animals were pooled at each indicated ZT between ZT0-20. Protein levels of CDK5, CRY1, and HSP90 were analyzed by western blot. (**C**) Diurnal activity of CDK5 was measured by an in vitro kinase assay. CDK5 was immunoprecipitated at each same time point between ZT0 and ZT20, and half of the immunoprecipitated material was used for performing an in vitro kinase assay using histone H1 (autoradiography, middle panel), whereas the other half was used to quantify the immunoprecipitated CDK5 (upper panel). Coomassie staining shows loading of the substrate (H1). Bottom panel: Quantification of three independent experiments (mean ± SEM). One-way ANOVA with Bonferroni's post-test, *: p<0.001. (**D**) The in vitro kinase assay was performed with SCN extracts at ZT12, and either LiCl (GSK3β inhibitor) or 34 µM roscovitine (CDK5 inhibitor). Histone H1 phosphorylation could not be detected with roscovitine treatment, showing the specificity of H1 phosphorylation by CDK5.

shRNAs against *Cdk5* in NIH 3T3 fibroblast cells (*Figure 2—figure supplement 1*) and subsequently injected into the SCN region adeno-associated viral particles containing expression vectors for either a scrambled set of shRNA or a *Cdk5*-specific shRNA (variant D, *Figure 2—figure supplement 1*). After recovery from the procedure the animals were transferred into cages containing a running-wheel in order to assess their activity profiles. The control animals expressing the scrambled set of

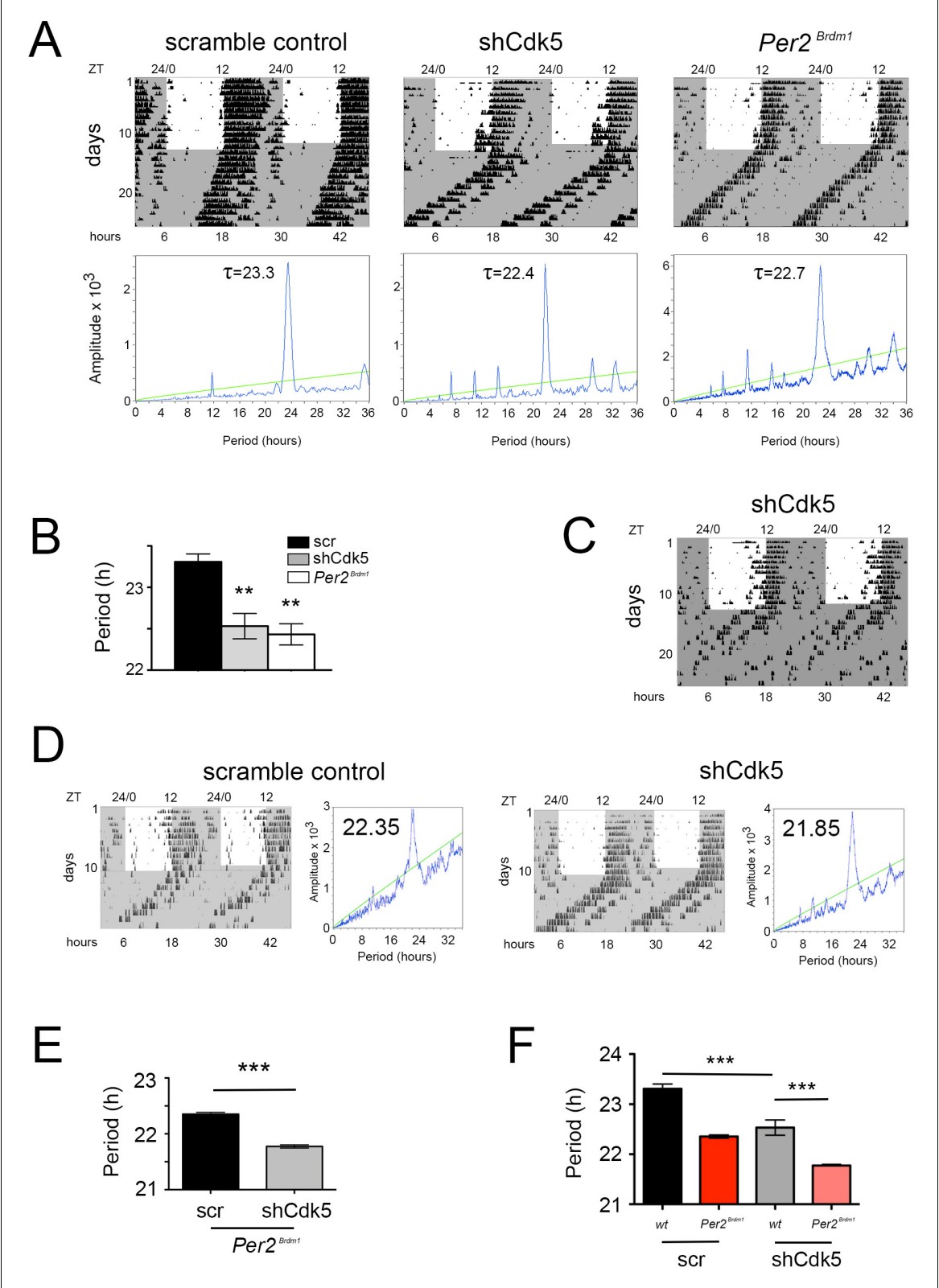

**Figure 2.** CDK5 affects the circadian clock. (**A**) Wheel-running activity of mice (black bins) infected with AAV expressing scrambled control shRNA, or shCdk5, and an animal with a deletion in the *Per2* gene (*Per2^Brdm1^*). The actograms are double plotted displaying in one row and below 2 consecutive days. The locomotor activity was confined to the dark period (shaded in gray), while under LD the mice displayed low activity during the light phase (white area). Under DD (continuous gray shaded area) the shCdk5 and *Per2^Brdm1^* animals show earlier onset of activity each day compared with the

*Figure 2 continued on next page*

*Figure 2 continued*

control animals. The $\chi^2$-periodogram analysis for each of the animals is shown below the corresponding actogram to determine the period length ($\tau$). (B) Quantification of the circadian period: 23.3 ± 0.1 hr for the control mice (n = 6, black bar), 22.5 ± 0.2 hr for shCdk5 injected mice (n = 6, gray bar), and 22.4 ± 0.1 hr for *Per2^Brdm1* mice (n = 4, white bar), (mean ± SEM). One-way ANOVA with Bonferroni's post-test, **p<0.01. (C) In some cases, mice in which *Cdk5* was silenced in the SCN became arrhythmic. (D) Wheel-running activity (black bins) of *Per2^Brdm1* mice infected with AAV expressing scrambled control shRNA (scr), or shRNA against Cdk5 (shCdk5). The actograms are double plotted displaying in one row and below 2 consecutive days. The dark shaded area indicates darkness during which the free-running period was determined. To the right of each actogram the corresponding $\chi^2$-periodogram is shown. The number in each periodogram indicates the period of the animal. (E) Quantification of the circadian period: 22.35 ± 0.03 hr for the scrambled *Per2^Brdm1* (n = 3, black bar) and 21.77 ± 0.03 hr for the shCdk5 injected *Per2^Brdm1* mice (n = 5, gray bar). Values are the mean ± SEM, t-test, ***p<0.0001. (F) 1-way ANOVA test on wild-type and *Per2^Brdm1* animals infected with AAV expressing scrambled control shRNA (scr), or shRNA against Cdk5 (shCdk5). N = 3–6 animals, error bars are the mean ± SEM, Bonferroni multiple comparisons test, ***p<0.001.

The online version of this article includes the following figure supplement(s) for figure 2:

**Figure supplement 1.** Characterization of shRNA against *Cdk5* Western blot using NIH 3T3 cell extracts transfected with different shRNAs against *Cdk5*.

**Figure supplement 2.** Additional activity plots of wild-type mice infected with AAV.

**Figure supplement 3.** Activity counts per day.

**Figure supplement 4.** Activity counts in dark or light phase.

**Figure supplement 5.** Additional activity plots of *Per2^Brdm1* mice infected with AAV.

---

shRNA displayed normal activity in the light-dark (LD) cycle with precise onset of activity at the beginning of the dark phase (ZT12). This onset of activity was less precise in mice with a *Cdk5* knock-down (shCdk5) but comparable to animals with a deletion mutation in the clock gene *Per2*, designated as *Per2^Brdm1* (**Figure 2A**, **Figure 2—figure supplement 2**). In constant darkness (DD), $\chi^2$-periodogram analysis revealed a normal average free-running period for the scramble control mice, whereas for shCdk5 and *Per2^Brdm1*, the period was significantly shortened (**Figure 2B**). In one case, the shCdk5 animals became arrhythmic (**Figure 2C**), again comparable to *Per2^Brdm1* mice that eventually became arrhythmic in DD as well (**Zheng et al., 1999**). The total wheel-running activity was significantly reduced in shCdk5 and *Per2^Brdm1* mice under DD as well as under LD conditions when compared with the scrambled control animals (**Figure 2—figure supplement 3**). The reduction of activity in the mutants under LD conditions is confined to the dark phase, but comparable between all three genotypes in the light phase (**Figure 2—figure supplement 4**). These results indicate that the period of the clock is affected by the lack of *Cdk5* expression in the SCN.

Interestingly, period in *Per2^Brdm1* mutant and wild-type shCdk5 knocked-down mice was not significantly different (**Figure 2B**), suggesting that CDK5 activity is linked to PER2 as indicated by our SDL screen (**Figure 1**). In order to test the contribution of *Cdk5*, we knocked down *Cdk5* in *Per2^Brdm1* mutant mice. This even further shortened period in *Per2^Brdm1* mutant animals compared to scramble control *Per2^Brdm1* animals (**Figure 2D,E**, **Figure 2—figure supplement 5**). This effect, however, was not simply additive ($\Delta$ wt versus wt KD ≈ 0.8 hr; $\Delta$ *Per2^Brdm1* versus *Per2^Brdm1* KD ≈ 0.6 hr, **Figure 2F**). Additionally, $\Delta$ wt KD versus *Per2^Brdm1* KD ≈ 0.8 hr, indicating that the difference in genetic background plays an important role. Overall, our observations suggest that Cdk5 may affect period partially via PER2 but also via additional factors (e.g. CLOCK, **Kwak et al., 2013**). Taken together, it appears that CDK5 is a main regulator of the circadian clock mechanism.

In order to confirm that the different phenotypes were associated with the accumulation levels of CDK5 in control and *Cdk5*-silenced mice, we performed immunofluorescence assays on coronal sections of the SCN. Sections were stained with DAPI (blue) in order to label nuclei, with GFP antibody (green) in order to show cells infected by the virus, and with CDK5 antibody (red) in order to compare protein accumulation between the two strains. Scramble as well as shCdk5 mice expressed GFP in the SCN, indicating that the two different viruses infected cells in this brain region (**Figure 3A**, **Figure 3—figure supplements 1–2**). The expression of *Cdk5* was efficiently suppressed in the SCN by the shCdk5 but not by the scrambled shRNA (**Figure 3A**, **Figure 3—figure supplements 1–2**), indicating that the behavioral phenotypes observed are due to efficient knock-down of *Cdk5*. The *Cdk5* shRNAs was expressed in the SCN (the injection site) and to some extent also

dorsal to the SCN but not in distant brain regions (i.e. the piriform cortex) as confirmed by lack of the GFP signal outside of the targeted region (*Figure 3A*).

Surprisingly, the phenotypes of shCdk5 and *Per2^Brdm1^* mice showed considerable similarity, implicating that the levels of PER2 accumulation might be similar in these two different mouse strains. In order to test whether *Cdk5* knock-down affected PER2, we stained with DAPI (blue) and immunostained with anti-PER2 (red) SCN sections obtained from control, shCdk5 and *Per2^Brdm1^* mice perfused at ZT12. PER2 was observed in the SCN of scramble controls, but was strongly reduced in shCdk5 and almost absent in *Per2^Brdm1^* animals (*Figure 3B*, *Figure 3—figure supplements 3–4*). These data suggested that CDK5 is a main regulator of the core circadian clock in the SCN and may alter PER2 accumulation and potentially other proteins involved in clock regulation.

## CDK5 interacts with PER2 protein in a temporal fashion

A study in *Drosophila* has shown that several kinases, including cyclin-dependent kinases, phosphorylate specific sites on per to maintain the circadian period (*Garbe et al., 2013*). Therefore, we aimed to understand whether a molecular interaction exists between CDK5 and PER2, as suggested by our SDL screen (*Figure 1*). We transfected cells with *Per2* and *Cdk5* expression vectors and tested whether the two proteins co-immunoprecipitated. We observed that immunoprecipitation with an anti-CDK5 antibody pulled down PER2 protein in two different cell lines (*Figure 4A*, *Figure 4—figure supplement 1*). Similar interactions were observed when cells were transfected with expression constructs resulting in PER2 and CDK5 proteins fused to short amino-acid tags of viral protein 5 (V5)

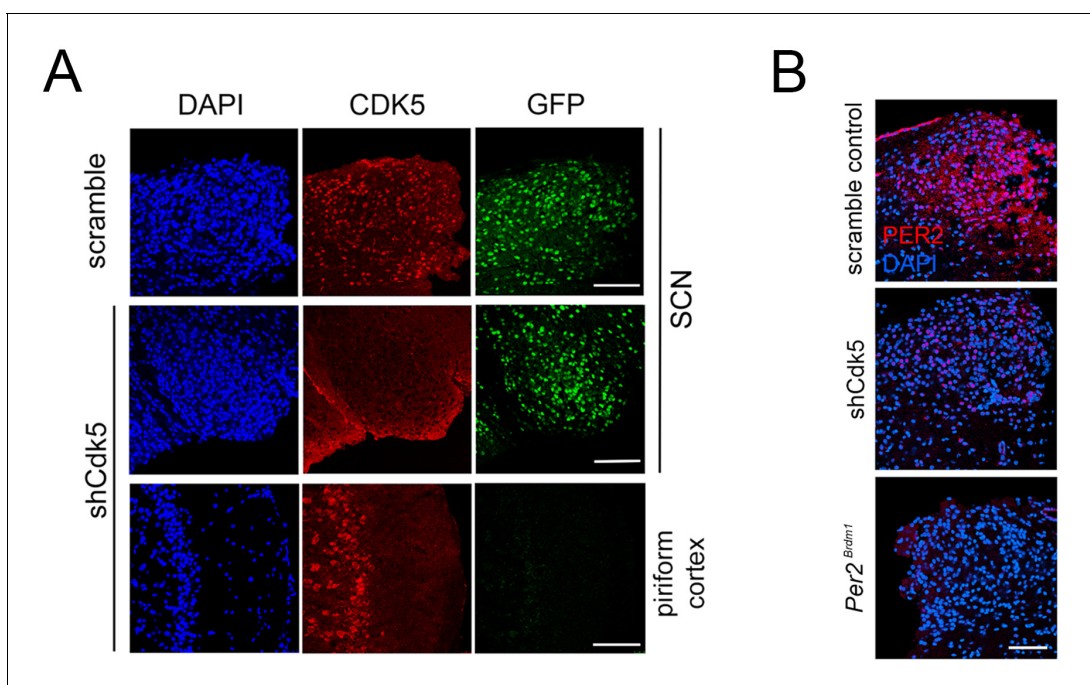

**Figure 3.** Immunohistochemistry in the SCN of control and shCdk5 silenced wild type and *Per2^Brdm1^* mice. (**A**) Representative sections of the SCN region after injection of AAVs carrying either scrambled shRNA, or shCdk5. Slices were stained with DAPI (blue), or anti-GFP (green) and anti-CDK5 (red) antibodies. GFP was used as marker for those cells infected by the virus. CDK5 was efficiently down-regulated in the SCN by shCdk5 (red panels) but not by scrambled shRNA, which was as efficiently delivered as shCDK5. As control, the non-infected piriform cortex from the same animal in which *Cdk5* was silenced is shown. Scale bar: 200 μm. (**B**) Analysis of PER2 expression in sections of the SCN of scrambled shRNA, shCdk5 and *Per2^Brdm1^* mice. Silencing of *Cdk5* leads to lack of PER2 (red) compared with control at ZT12, which almost resembles the situation observed in *Per2^Brdm1^* animals. Blue color: DAPI staining for cell nuclei. Scale bar: 200 μm.

The online version of this article includes the following figure supplement(s) for figure 3:

**Figure supplement 1.** Lower magnification of SCN sections stained for CDK5.
**Figure supplement 2.** Quantification of CDK5 signal.
**Figure supplement 3.** Lower magnification of SCN sections stained for PER2.
**Figure supplement 4.** Quantification of PER2 signal.

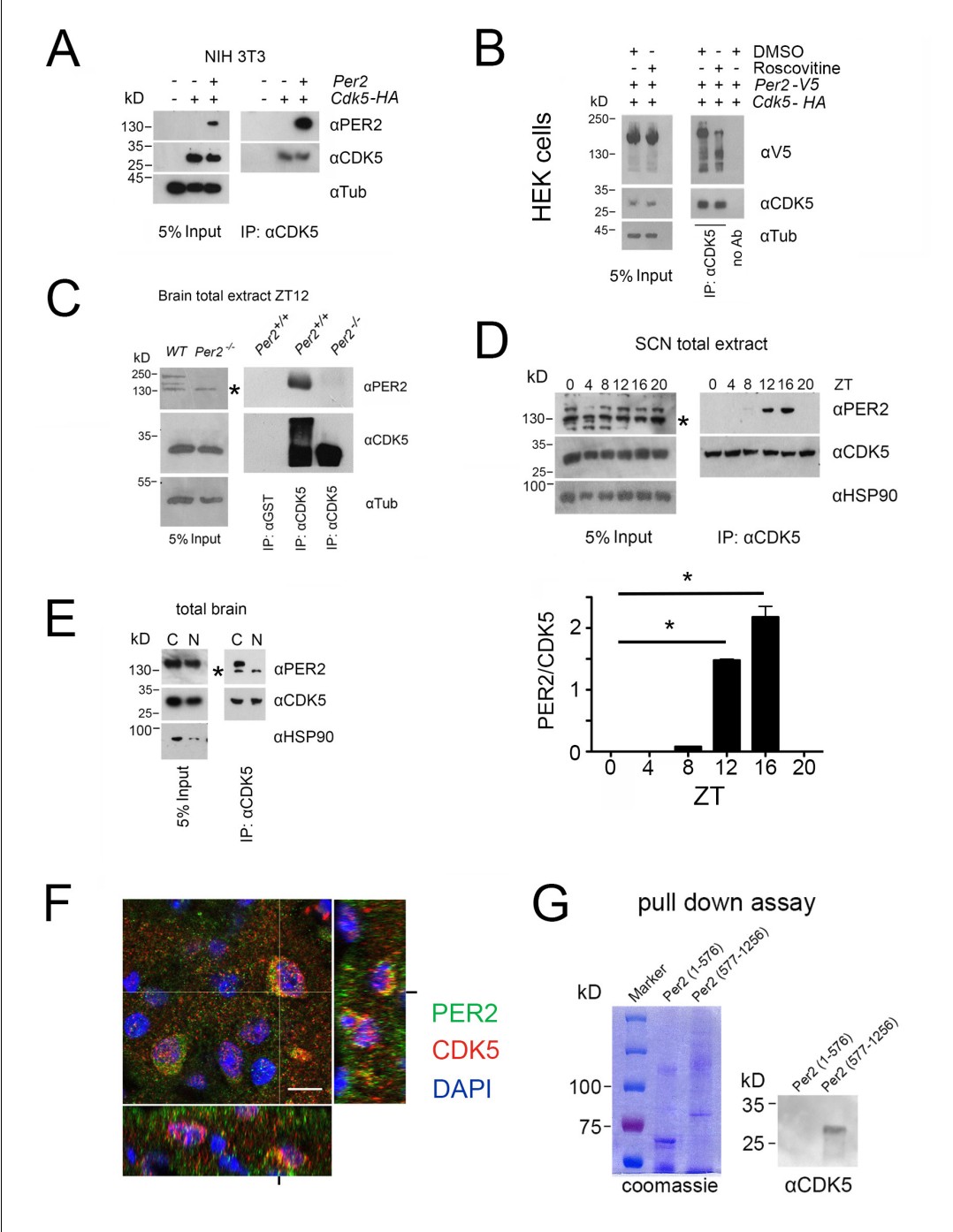

**Figure 4.** PER2 interacts with CDK5 in a temporal fashion in the cytoplasm. (**A**) Overexpression of PER2 and CDK5-HA in NIH 3T3 cells and subsequent immunoprecipitation (IP) using an anti-CDK5 antibody. The left panel shows 5% of the input and the right panel co-precipitation of PER2 with CDK5. (**B**) Overexpression of PER2-V5 and CDK5-HA in HEK293 cells in presence or absence of 34 μM roscovitine (CDK5 inhibitor) and DMSO (solvent). Left panel shows 5% of the input and the right panel the immunoprecipitation with anti-CDK5 or without antibody. (**C**) Immunoprecipitation (IP) of PER2 and CDK5 from total mouse brain extract collected at ZT12. Left panel shows the input. The right panel depicts co-immunoprecipitation of PER2 and CDK5 using either anti-CDK5 antibody or anti-GST antibody for precipitation. The middle lane shows PER2-CDK5 co-immunoprecipitation in control animals (*Per2*⁺/⁺) but not in *Per2*⁻/⁻ mice illustrating the specificity of the PER2-CDK5 interaction. The * in the blot indicates unspecific signal. (**D**) Temporal profile of the PER2-CDK5 interaction in total extracts from SCN tissue around the clock. Input was analyzed by immunoblot using anti-CDK5, anti-PER2, and anti-HSP90 antibodies (left panel). CDK5 co-immunoprecipitated PER2 in a diurnal fashion with a peak between ZT12 and ZT16. The statistical analysis of the PER2/CDK5 signal around the clock is shown below (one-way ANOVA with Bonferroni's post-test, n = 3, *p<0.0001, values are mean ± SEM). * in the blot indicates unspecific signal. (**E**) Immunoprecipitation of PER2 with CDK5 from cytoplasmic and nuclear brain extracts collected at ZT12. The left

*Figure 4 continued on next page*

*Figure 4 continued*

panel shows the input and the right panel co-IP of PER2 and CDK5, which occurs only in the cytoplasm but not in the nucleus. The smaller band detected by the anti-PER2 antibody depicts an unspecific band that is smaller than PER2. * in the blot indicates unspecific signal. (F) Slices from the SCN obtained at ZT12 were immunostained with PER2 antibody (green), CDK5 (red), and nuclei were marked with DAPI (blue). Co-localization of the two proteins results in the yellow color. Scale bar: 10 μm. The z-stacks right and below the micrograph confirm co-localization of PER2 and CDK5 (yellow). (G) Purification of the N-terminal half of PER2 (1–576) or the C-terminal half (577–1256) (left panel, coomassie). CDK5-His was pulled down by both recombinant PER2 attached to the glutathione resin, but only the C-terminal was able to retain CDK5 (immunoblot using anti-His antibody, right panel).

The online version of this article includes the following figure supplement(s) for figure 4:

**Figure supplement 1.** PER2-CDK5 interaction in HEK cells.
**Figure supplement 2.** PER2-CDK5 interaction at different salt concentrations.
**Figure supplement 3.** Co-localization of PER2 and CDK5 in SCN tissue.
**Figure supplement 4.** Deletion of PAS domains had no influence on PER2-CDK5 interaction.
**Figure supplement 5.** Scheme of PER2 fragments used for the pull-down assy.

and haemaglutinine (HA) fused to them, respectively (*Figure 4B*). Interestingly, interaction between PER2-V5 and CDK5-HA was reduced when roscovitine, which inhibits interaction of CDK5 with its targets (*Hsu et al., 2013*), was added to the cells (*Figure 4B*). This suggested that active CDK5 protein interacted better with PER2 than CDK5 in its inhibited form.

In order to test whether this interaction could be observed in tissue, we prepared total brain extracts at ZT12, when kinase activity of CDK5 was high (*Figure 1C*). At two different salt concentrations, we could pull-down PER2 and CDK5 using either anti-CDK5 or anti-PER2 antibodies (*Figure 4—figure supplement 2*). The specificity of the signals was confirmed by using brain extracts from *Per2⁻/⁻* mice (*Chavan et al., 2016*) that completely lack PER2 protein (*Figure 4C*). Next, we wanted to investigate whether the interaction between the two proteins is time of day-dependent in the SCN. Total extracts of SCN tissue at ZT0, 4, 8, 12, 16 and 20 were prepared and immunoprecipitation with an anti-CDK5 antibody pulled down PER2 at ZT8, 12, and 16, with the strongest signals at ZT12 and ZT16 (*Figure 4D*). Taken together, these observations suggested that the interaction between CDK5 and PER2 can occur in brain tissue and that in the SCN this interaction was time of day-dependent. This observation was confirmed on SCN tissue sections, where we observed PER2 expression at ZT12 but less at ZT0 with co-localization of CDK5 restricted to ZT12 (*Figure 4—figure supplement 3*).

Next, we tested in which subcellular compartment the interaction between CDK5 and PER2 takes place. We prepared nuclear and cytoplasmic extracts from total brain tissue and performed immunoprecipitation using an anti-CDK5 antibody. PER2 could only be observed in the cytoplasmic but not the nuclear fraction (*Figure 4E*). This was supported by the observation that the two proteins were co-localized only in the cytoplasm in SCN tissue (*Figure 4F*, yellow color).

Furthermore, we evaluated with which part of PER2 the CDK5 protein interacts. We tested whether deletions in the PAS-domain of PER2, a known domain for protein interactions (*Ponting and Aravind, 1997*), influenced CDK5 binding. No significant effect of deletions of the PAS-A and PAS-B domains on the interaction was observed (*Figure 4—figure supplement 4*). Next, we generated expression vectors coding either for the N-terminal (1-576) or the C-terminal part (577–1257) of PER2 fused to GST (*Figure 4—figure supplement 5*). The recombinant forms of PER2 and histidine-tagged CDK5 were produced in bacteria. A pull-down assay with these proteins showed that the C-terminal but not the N-terminal half of the PER2 protein was pulled-down by CDK5, suggesting that CDK5 binds to the C-terminal part of PER2 (*Figure 4G*). This does, however, not exclude weak interactions of the CDK5 protein with the N-terminal half in vivo. Taken together, our data suggest a physical interaction of PER2 and CDK5 in the cytoplasm.

## CDK5 phosphorylates PER2 at serine 394

In order to understand whether CDK5 phosphorylates the PER2 protein we overexpressed the N-terminal and C-terminal parts of PER2 fused to GST in bacteria (*Figure 5—figure supplement 1*) and performed an in vitro kinase assay with the recombinant proteins. Recombinant CDK5/p35 protein complex along with $\gamma$-$^{32}$P labeled ATP resulted in phosphorylation of the N-terminal part of the PER2 protein with a main signal at around 120 kD (*Figure 5A*, *Figure 5—figure supplement 2*, $^{32}$P

panels). Addition of roscovitine abolished phosphorylation of PER2 whereas LiCl had no effect (*Figure 5—figure supplement 3*). Interestingly, no phosphorylation of the C-terminal part of PER2 was observed, only a signal corresponding to the auto-phosphorylation of CDK5 was detected at around 60 kD (*Figure 5A*, $^{32}$P panel).

Next, we aimed to identify the phosphorylation site(s) in the N-terminal part of PER2 using the recombinant protein, which was phosphorylated by CDK5/p35 in vitro. Mass spectrometry revealed several phosphorylation sites at serine and threonine residues, respectively (*Supplementary file 1*). One of the serine residues of PER2 was located within a CDK5 consensus sequence and had the highest probability score for being phosphorylated (*Figure 5B*). The serine residue at position 394 (S394) of PER2 is located at the end of the PAS domain and within the deletion of the mutated PER2 of *Per2^{Brdm1}* mice (*Zheng et al., 1999*). This suggested that CDK5/p35 phosphorylates S394 and that this phosphorylation is of functional relevance. Mutations of this serine to aspartic acid (S394D) or glycine (S394G) reduced phosphorylation by CDK5/p35 significantly (*Figure 5C*), confirming that CDK5/p35 phosphorylated S394. Next, we produced a monoclonal antibody against the phosphorylated serine at 394 of PER2 (P-S394-PER2) (*Figure 5—figure supplements 4–6*). With this antibody we detected the phosphorylated N-terminal fragment of PER2 in presence of CDK5/p35 but not when S394 was mutated to glycine (S394G) or when CDK5 was inhibited by roscovitine (*Figure 5D*), confirming S394 phosphorylation by CDK5/p35.

In order to determine whether PER2 phosphorylation at S394 is time of day-dependent, we collected SCN tissue every 4 hr. The P-S394-PER2 specific antibody detected highest phosphorylation at ZT12 with weaker or no phosphorylation at other time points indicating that S394 is phosphorylated in a time of day-dependent manner (*Figure 5E*). Fractionation of wild-type brain cellular extracts prepared at ZT12 into nuclear and cytoplasmic parts showed phosphorylated S394 predominantly in the cytoplasm with little or no signal in the nucleus when labeled with the P-S394-PER2 antibody (*Figure 5F*). Total PER2 was observed in both cellular compartments with higher levels in the nucleus (*Figure 5F*). This suggested that phosphorylation of S394 of PER2 happens predominantly in the cytoplasm and that this phosphorylation is either removed or occluded when PER2 enters the nucleus.

## CDK5 affects stability and nuclear localization of PER2

To evaluate the function of CDK5-driven PER2 phosphorylation, we wanted to determine whether CDK5 affects PER2 stability. We treated NIH 3T3 cells with roscovitine and DMSO as control and determined endogenous levels of PER2. We observed that roscovitine treatment of cells reduced PER2 levels, suggesting that CDK5 can affect protein stability (*Figure 6A*). In order to challenge this observation, we deleted *Cdk5* in NIH 3T3 cells using the CRISPR/Cas9 method (*Figure 6—figure supplements 1–3*). We observed that deletion of *Cdk5* led to reduced amounts of PER2 (*Figure 6B*), consistent with the data shown in *Figure 6A*. These observations support the notion that phosphorylation by CDK5 affects PER2 abundance. In order to monitor PER2 stability, we knocked down *Cdk5* using the shRNA D (*Figure 2—figure supplement 1*). We observed that increasing amounts of shCdk5 dampened PER2 levels proportionally to the decreasing CDK5 levels (*Figure 6C*).

In order to determine whether CDK5 modulates degradation of PER2, we blocked protein synthesis using cycloheximide. Under conditions that partially knocked down *Cdk5* (at a concentration of 2.7 µM of shCdk5, *Figure 6C*), we measured PER2 and CDK5 protein levels over 6 hr after cycloheximide treatment. We found that degradation of PER2 was faster when *Cdk5* was knocked down compared with unspecific shRNA treatment (shCdk5 $t_{1/2}$=4 h, scr $t_{1/2}$=11 h) (*Figure 6D*), indicating that reduction of *Cdk5* accelerated PER2 degradation. Next, we investigated whether PER2 degradation involved the proteasome. Cells were treated with epoxomycin, a proteasome inhibitor, or with the solvent DMSO. In line with our previous experiments, shCdk5 treatment efficiently knocked down CDK5 and reduced PER2 levels compared with scrambled shRNA treatment. Addition of epoxomycin, but not DMSO, significantly increased PER2 levels despite absence of CDK5 (*Figure 6E*), indicating that PER2 degradation involved the proteasome. Residual amounts of CDK5 in the cells still may phosphorylate PER2 and direct it into the nucleus. Therefore, we wanted to see whether PER2 could be detected in nuclear extracts of shCdk5 knocked down cells. In line with our previous observations we did not detect PER2 in nuclear extract (*Figure 6F*), supporting the idea that PER2 needed to be phosphorylated by CDK5 in order to enter the nucleus. Data from immunofluorescence experiments

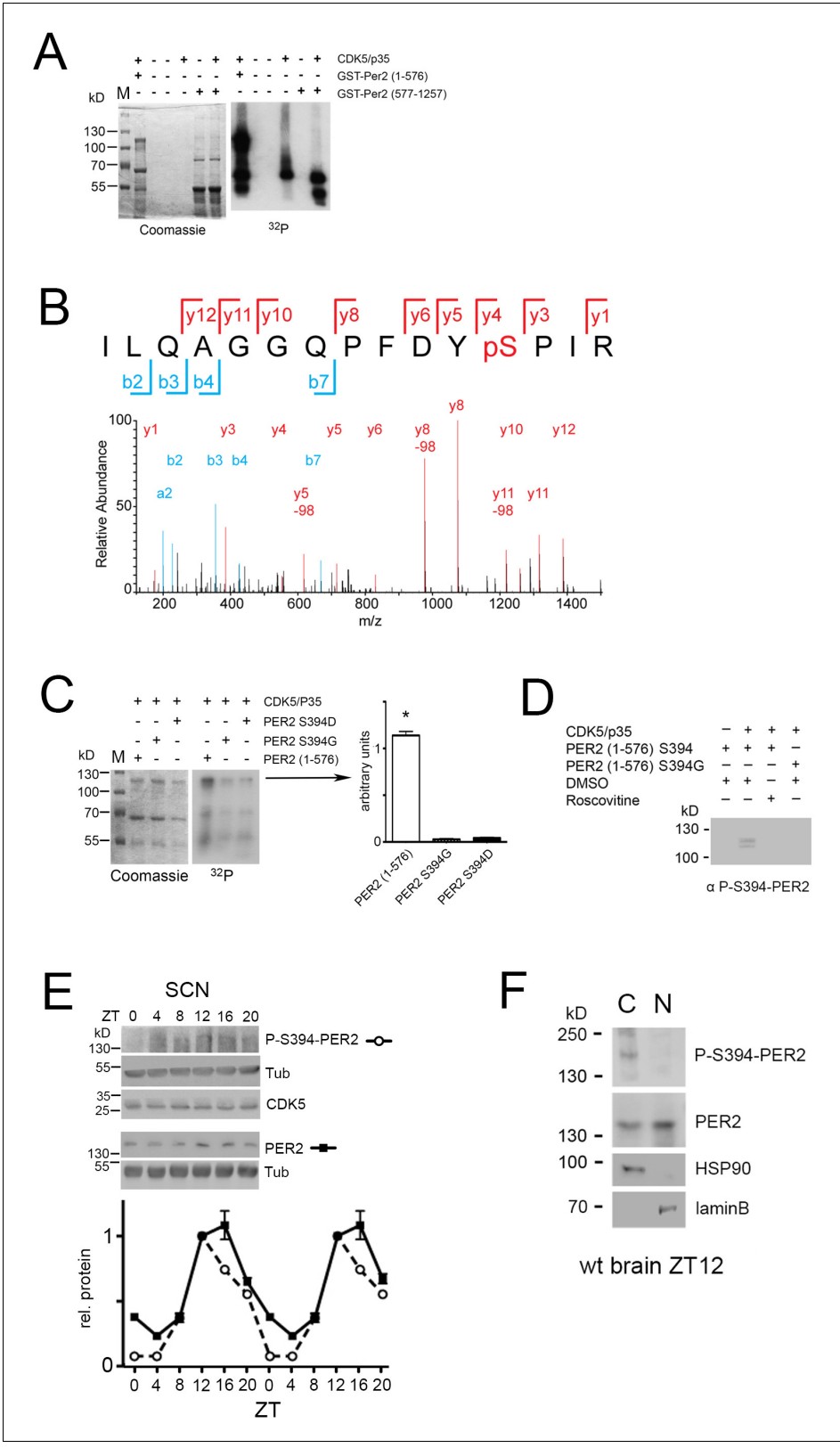

**Figure 5.** CDK5 phosphorylates PER2 at S394. (**A**) An in vitro kinase assay was performed using recombinant CDK5/p35 and either GST-PER2 1–576 or GST-PER2 577–1256 as substrate. The samples were subjected to 10% SDS page (Coomassie, left panel) and the phosphorylation of PER2 was detected by autoradiography in order to visualize $^{32}$P-labeled proteins (right panel). CDK5 phosphorylates the N-terminal half (1-576) of a GST-PER2 fusion

*Figure 5 continued on next page*

*Figure 5 continued*

protein whereas the C-terminal half (577–1257) is not phosphorylated. The signal for CDK5/p35 alone indicates CDK5 auto-phosphorylation seen in all lanes when CDK5 is present. (**B**) Annotated mass spectrum of the tryptic peptide PER2[383-397] ILQAGGQPFDYpSPIR containing the phosphorylated residue S394. The red color depicts the y-ion series (1-12) and blue the b-ion series (2–7, a2); y5-98, y8-98, y11-98 show the de-phosphorylated ions. (**C**) In vitro kinase assay was performed as in (**A**). The putative phosphorylation site was mutated to aspartic acid (S394D) or glycine (S394G). Both mutations abrogated the CDK5-mediated phosphorylation. Coomassie staining reveals equal expression of the GST-PER2 fragments. The bar diagram at the right shows the quantification of three experiments. One-way-ANOVA with Bonferroni's post-test, *: p<0.001 (**D**) The monoclonal antibody produced against P-S394-PER2 does recognizes PER2 (1–576) S394 phosphorylation mediated by CDK5/p35 in presence but not in absence of the kinase or when CDK5 is inactivated by roscovitine. This antibody does not recognize the S394G mutated form even in presence of CDK5/p35. (**E**) Temporal profile of P-S394-PER2 and total PER2 in SCN tissue. Upper panels show western blots of the corresponding proteins indicated on the right. Below the quantification of three experiments is shown, in which the value at ZT12 of PER2 has been set to 1. The data were double plotted. Values are the mean ± SEM. Two-way ANOVA with Bonferroni's multiple comparisons revealed that the two curves are significantly different with p<0.0001, F = 93.65, DFn = 1, DFd = 48. (**F**) Subcellular localization of P-S394-PER2. Total wild-type mouse brain extracts were separated into cytoplasmic (HSP90 positive) and nuclear (laminB positive) fractions. Phosphorylated PER2 was predominantly detected in the cytoplasm with the P-S394-PER2 antibody, whereas the general PER2 antibody detected PER2 in both compartments with higher amounts in the nuclear fraction.

The online version of this article includes the following figure supplement(s) for figure 5:

**Figure supplement 1.** Scheme of PER2 fragments used for the in vitro kinase assay.
**Figure supplement 2.** Additional controls for in vitro kinase assay.
**Figure supplement 3.** Testing specificity of the in vitro kinase assay.
**Figure supplement 4.** Characterization of antisera against P-S394-PER2.
**Figure supplement 5.** Characterization of hybridomas against P-S394-PER2.
**Figure supplement 6.** Validation of anti-P-S394-PER2 antibody.

on SCN sections were in line with this hypothesis. PER2 was only detected in nuclei when CDK5 was available (*Figure 6G*, arrowheads, *Figure 6—figure supplement 4*), but not when shCdk5 was expressed in SCN cells (*Figure 6G*, white arrow, *Figure 6—figure supplement 4*).

It has been described that nuclear entry of PER2 involves CRY1 (*Kume et al., 1999*; *Ollinger et al., 2014*). In addition, CRY1-mediated hetero-dimerization stabilizes PER2 by inhibiting its own ubiquitination (*Yagita et al., 2000*). Therefore, we tested the interaction potential of wild-type PER2 and the S394G PER2 mutation with CRY1 by overexpressing the two PER variants in NIH 3T3 cells. Immunoprecipitation of wild-type PER2 pulled down CRY1; however, the S394G PER2 mutation was significantly less efficient in doing so (*Figure 6H*). The small amounts of CRY1 detected may be bound to endogenous PER2 that is present in the cells. In summary, these experiments suggested that CDK5 affects PER2 stability, interaction with CRY1, and nuclear localization.

## Discussion

Not only do kinases play a crucial role in signal transduction in response to extracellular stimuli, but they also regulate cycling processes such as the cell cycle and circadian rhythms. Most cyclin dependent kinases (CDKs) regulate the cell cycle, with few exceptions such as the cyclin dependent kinase 5 (CDK5). This kinase is ubiquitously expressed and its function is vital in post-mitotic neurons, where other CDKs are not active. Although CDK5 is not implicated in cell cycle progression, it can aberrantly activate components of the cell cycle when it is dysregulated in post-mitotic neurons, leading to cell death (*Chang et al., 2012*). Interestingly, cell death is affected by the clock component PER2 as well (*Magnone et al., 2014*), suggesting that both, CDK5 and PER2 act in the same pathway, or that their pathways cross at a critical point during the regulation of cell death. The synthetic dosage lethal screen that we performed in yeast supports this notion, as expression of PER2 in yeast lacking *Cdk5* strongly and significantly compromised growth (*Figure 1A*).

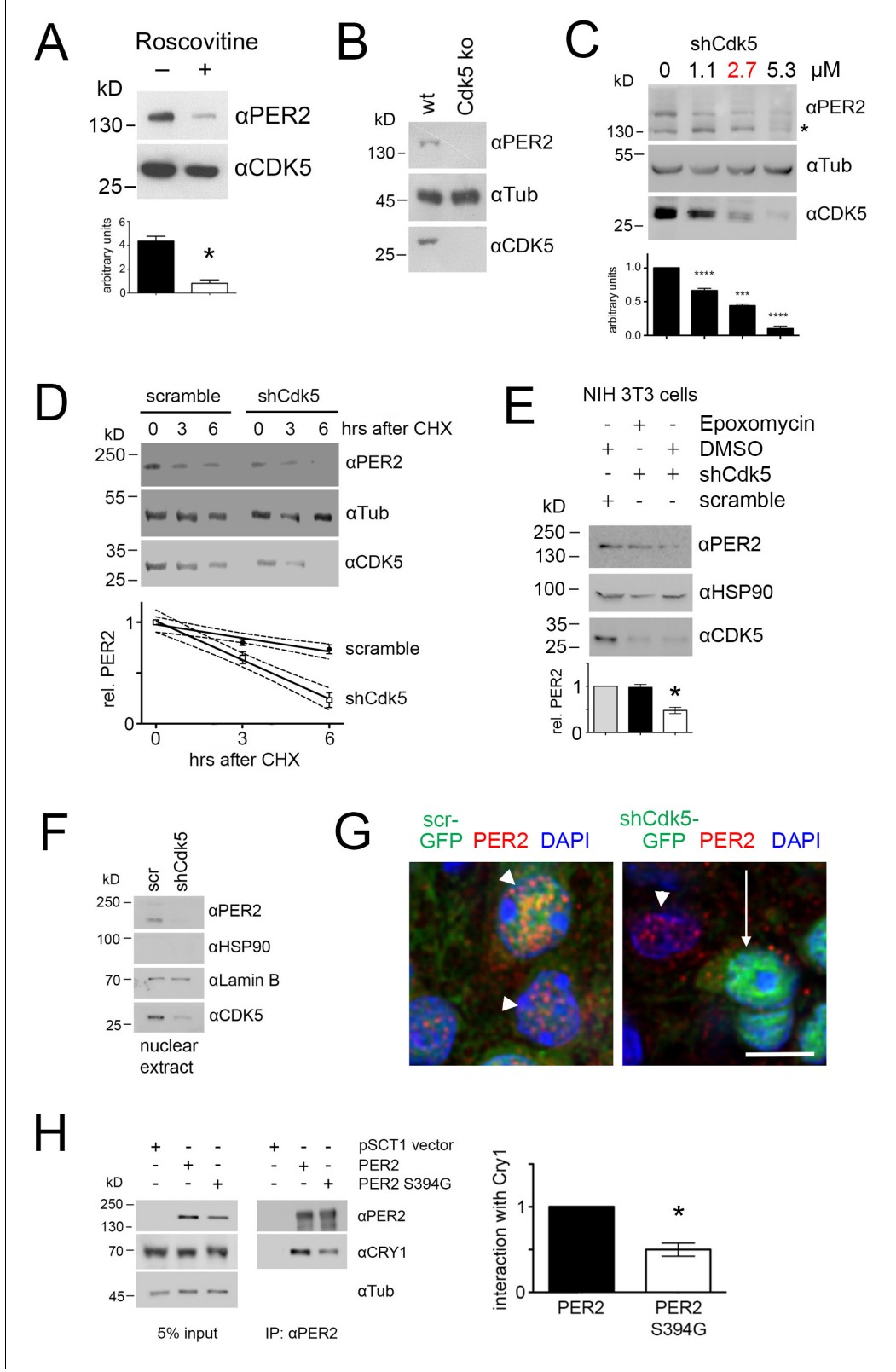

**Figure 6.** CDK5 affects PER2 stability and nuclear localization. (**A**) Western blot of NIH 3T3 cell extracts with and without roscovitine treatment. When roscovitine inhibited CDK5, less PER2 protein was detected in cell extracts. The bar diagram below shows values (mean ± SEM) of three experiments with significant differences between roscovitine treated and untreated cells, t-test, *p<0.001. (**B**) CRISPR/Cas9-mediated knockout of *Cdk5* in NIH 3T3

*Figure 6 continued on next page*

*Figure 6 continued*

cells. Western blot shows absence of PER2 in cells when *Cdk5* is deleted. (**C**) Titration of CDK5 knock-down as revealed by Western blotting. PER2 levels decreased proportionally to increasing amounts of shCdk5. 2.7 µM of shCdk5 (red) was used for subsequent experiments. The value without shCdk5 was set to 1. One-way ANOVA with Bonferroni post-test, n = 4, \*\*\*p<0.001, \*\*\*\*p<0.0001, mean ± SD. The \* in the blot indicates unspecific signal. (**D**) Temporal profile of protein abundance in NIH 3T3 cells 0, 3 and 6 hr after inhibition of protein synthesis by 100 µM cycloheximide (CHX) in presence of scrambled shRNA, or shCdk5, respectively (2.7 µM of the respective shRNA was used). The diagram below shows quantification of PER2 protein over time. Linear regression with 95% confidence intervals (hatched lines) indicates that knock-down of *Cdk5* leads to less stable PER2 (shCdk5 $t_{1/2}$=4h, scr $t_{1/2}$=11h). Two-way ANOVA with Bonferroni's post-test revealed that the two curves are significantly different, n = 3, p<0.01, F = 24.53, DFn = 1, DFd = 4. (**E**) Inhibition of the proteasome by epoxomycin in cells with shCdk5 leads to amounts of PER2 that are higher compared with the levels without epoxomycin treatment and are comparable to the levels observed in cells without *Cdk5* knockdown. Diagram below displays the quantification of three experiments. Scrambled shRNA values were set to 1. One-way ANOVA with Bonferroni's post-test shows no significant reduction of PER2 in shCdk5 cells in presence of epoxomycin, but significantly lower values in absence of epoxomycin when compared with scrambled shRNA treatment. One-way ANOVA with Bonferroni's post-test, n = 3, p<0.001. (**F**) PER2 abundance in nuclear extracts of NIH 3T3 cells. Knockdown of *Cdk5* reduces PER2 levels in the nucleus as revealed by Western blotting. HSP90 = cytosolic marker, LaminB = nuclear maker. (**G**) Immunofluorescence of PER2 (red) at ZT12 in mouse SCN sections after infection with AAV (green) expressing scrambled shRNA (left panel), or shCdk5 (right panel). Nuclei are visualized by DAPI staining (blue). PER2 can only be observed in the nucleus in presence (white arrow heads) but not in absence of CDK5 (white arrow). Scale bar = 7.5 µm. (**H**) Co-immunoprecipitation of CRY1 by PER2 in NIH 3T3 cells. Substitution of S394 to G in PER2 reduces the levels of co-precipitated CRY1 (right panel). The left panel shows the input. The bar diagram on the right displays the quantification of three experiments, where the amount of precipitated CRY1 by PER2 is set to 1. Paired t-test reveals a significant difference between the amounts of CRY1 precipitated by PER2 and the S394G PER2 mutation, n = 3, \*p<0.05, mean ± SD.

The online version of this article includes the following figure supplement(s) for figure 6:

**Figure supplement 1.** Characterization of *Cdk5* ko cell morphology.
**Figure supplement 2.** Selection for absence of *Cdk5* mRNA.
**Figure supplement 3.** Selection for absence of CDK5 protein.
**Figure supplement 4.** Additional examples of cellular localization of PER2.

The kinase CDK5 displays many effects that ensure proper brain function and development. Mice deficient for *Cdk5* are perinatal lethal (*Gilmore et al., 1998*; *Ohshima et al., 1996*). CDK5 influences cortical neuron migration, cerebellar development, synapse formation and plasticity (*Kawauchi, 2014*). Here, we identified a new role for this kinase, that is the regulation of the circadian clock in vivo. Previously, CDK5 had been identified to phosphorylate CLOCK and thereby regulate CLOCK stability and cellular distribution in cells (*Kwak et al., 2013*). In the SCN, however, NPAS2 may replace the function of CLOCK (*Debruyne et al., 2006*; *DeBruyne et al., 2007*) and therefore phosphorylation of CLOCK by CDK5 may play a minor role in the SCN. Hence, to unravel the novel function of CDK5 in the circadian oscillator, we had to restrict ourselves to the use of SCN tissue and whole animals.

CDK5 activity, but not its protein accumulation, displays a diurnal profile in the SCN with high activity during the night and low activity during the day (*Figure 1C*). The activity displayed a typical on/off profile similar to other CDKs. This finding raises the question how this diurnal activity of CDK5 may be achieved. On one hand, ATP accumulation, which is required for phosphorylation, peaks during the night in the SCN (*Yamazaki et al., 1994*). On the other hand, CDK5 activity is regulated by cofactors. Depending on its cofactor, CDK5 in the brain phosphorylates targets involved in neurodegenerative diseases (e.g. Tau, MAP1B), neuronal migration (e.g. DCX), and synaptic signaling (e.g. $Ca_v2.2$, Dynamin1, NR2A, DARPP-32) (*Kawauchi, 2014*). The most obvious candidates to regulate its time-dependent activity would be cyclins D1 and E, which inhibit CDK5, or cyclin I, which activates it. Alternatively, other known CDK5 regulators such as p35 may be involved (*Shah and Lahiri, 2014*). Most likely, positive and negative feedback loops of other kinases and phosphatases are necessary to generate the on/off profile, although the components involved in this mechanism are probably

different from the ones known for CDKs that regulate the cell cycle. Interestingly, CK1 phosphorylates and activates CDK5 in vitro (*Sharma et al., 1999*) and CDK5 is thought to phosphorylate and inhibit CK1δ in vitro (*Ianes et al., 2016*; *Eng et al., 2017*) potentially establishing a feedback loop between the two kinases. However, additional research is needed to determine the precise mechanism of diurnal on/off activation of CDK5.

As evidenced in *Figure 2*, Cdk5 knock-down affects circadian clock period at the behavioral level. The shortening of period in mice with knocked-down Cdk5 is comparable to mice containing a mutation of the *Per2* gene (*Per2*$^{Brdm1}$ mutant mice, *Zheng et al., 1999*). Interestingly, however, knock-down of Cdk5 in *Per2*$^{Brdm1}$ mutant mice leads to further shortening of circadian period. This suggests that Cdk5 may affect period either independently of *Per2* or, while PER2 may still be important, CDK5 regulates other proteins important for clock function. Since the difference between the period in control versus Cdk5 knock-down (0.8 hr, black and gray bars, *Figure 2F*) is not the same as in *Per2*$^{Brdm1}$ mutant versus its Cdk5 knock-down (0.6 hr, red and rose bars, *Figure 2F*) the second possibility is more likely. Moreover, wt KD versus *Per2*$^{Brdm1}$ KD show a difference in period (*Figure 2F*, gray and rose bars), suggesting that the difference in genotype plays an important role. From a dynamic perspective, it is possible that lack of PER2 protein will unmask Cdk5 targets that otherwise would be phosphorylated less efficiently or not at all. For example, the PER2 site that is phosphorylated by CDK5 (PFDY<u>S</u>PIR) is very similar in PER1 (PFDH<u>S</u>PIR). If PER1 would be phosphorylated by CDK5 at this site at the same rate as PER2, then PER1 as well as PER2 should be absent in the nucleus of SCN cells. This would correspond to a PER1/PER2 double knock-out, which become immediately arrhythmic when subjected to constant darkness (*Zheng et al., 2001*). This is not the phenotype we observe in the Cdk5 knock-down mice and hence it is unlikely that CDK5 affects PER1 in the same manner as it affects PER2. However, in the absence of PER2 the dynamics may change and PER1 may become a better target for CDK5 and influence period. This view is consistent with the observation that knock-down of Cdk5 in *Per2*$^{Brdm1}$ mutant mice can shorten period (*Figure 2D, E*).

CDK5 binds to the C-terminal half of PER2 (*Figure 4G*) and phosphorylates it at S394 (*Figure 5*), which is located in the PAC domain of the N-terminal half of the protein. Hence, the binding and phosphorylation sites are far apart, suggesting a structure of PER2 allowing proximity of the CDK5 binding and phosphorylation domains. We cannot exclude weak binding of CDK5 to the N-terminal half of PER2, because phosphorylation at S394 occurs in vitro even in the absence of the C-terminal half of the PER2 protein (*Figure 5A*). This may be due to the fact that the N-terminal half is overexpressed in vitro, which strongly increases the probability of phosphorylation by CDK5 even in the absence of the C-terminal binding domain. It is also known that p35 (which is used in the in vitro kinase assay to activate CDK5) can increase the interaction between CDK5 and its targets (*Hsu et al., 2013*).

In SCN tissue PER2 phosphorylation at S394 appears to be time of day-dependent, with highest levels at ZT12 and ZT16 (*Figure 5E*) when CDK5 activity is high (*Figure 1C*). Compared with total PER2 protein the S394 phosphorylated form appears to be slightly advanced in its phase. The difference in phase is probably even larger than it appears here, because the polyclonal antibody that detects total PER2 also detects the phosphorylated S394 PER2 variant. This is especially important in the rise of the signal detected, which appears to be identical in *Figure 5E*. Probably the steep increase between ZT8 and ZT12 represents the S394 phosphorylated forms in both curves. In contrast, the decrease in PER2 levels differs between total PER2 and P-S394-PER2 form. Consistent with previous studies total PER2 peaks in the nucleus at ZT16 in the SCN (*Nam et al., 2014*) when P-S394-PER2 is not detected anymore. This highlights that additional post-translational modifications of PER2 exist (*Toh et al., 2001*; *Vanselow et al., 2006*) and that P-S394-PER2 disappears faster compared with other modified forms. Probably, P-S394-PER2 plays a role in PER2 dynamics in terms of shuttling from the cytoplasm to the nucleus, because P-S394-PER2 can only be observed in the cytoplasmic and not the nuclear fraction (*Figure 5F*). The phosphorylation of PER2 by CDK5 may therefore be critical for the assembly of a macromolecular complex in the cytoplasm (*Aryal et al., 2017*), which then enters the nucleus.

The difference in the decline between PER2 and its S394 phosphorylated form in the SCN may suggest a role of the S394 phosphorylation not only for nuclear transport but also for PER2 protein stability. The earlier decline of the P-S394-PER2 signal compared with total PER2 (*Figure 5F*) might suggest that the S394 phosphorylated form is less stable. Apparently, the opposite is the case, as shown in

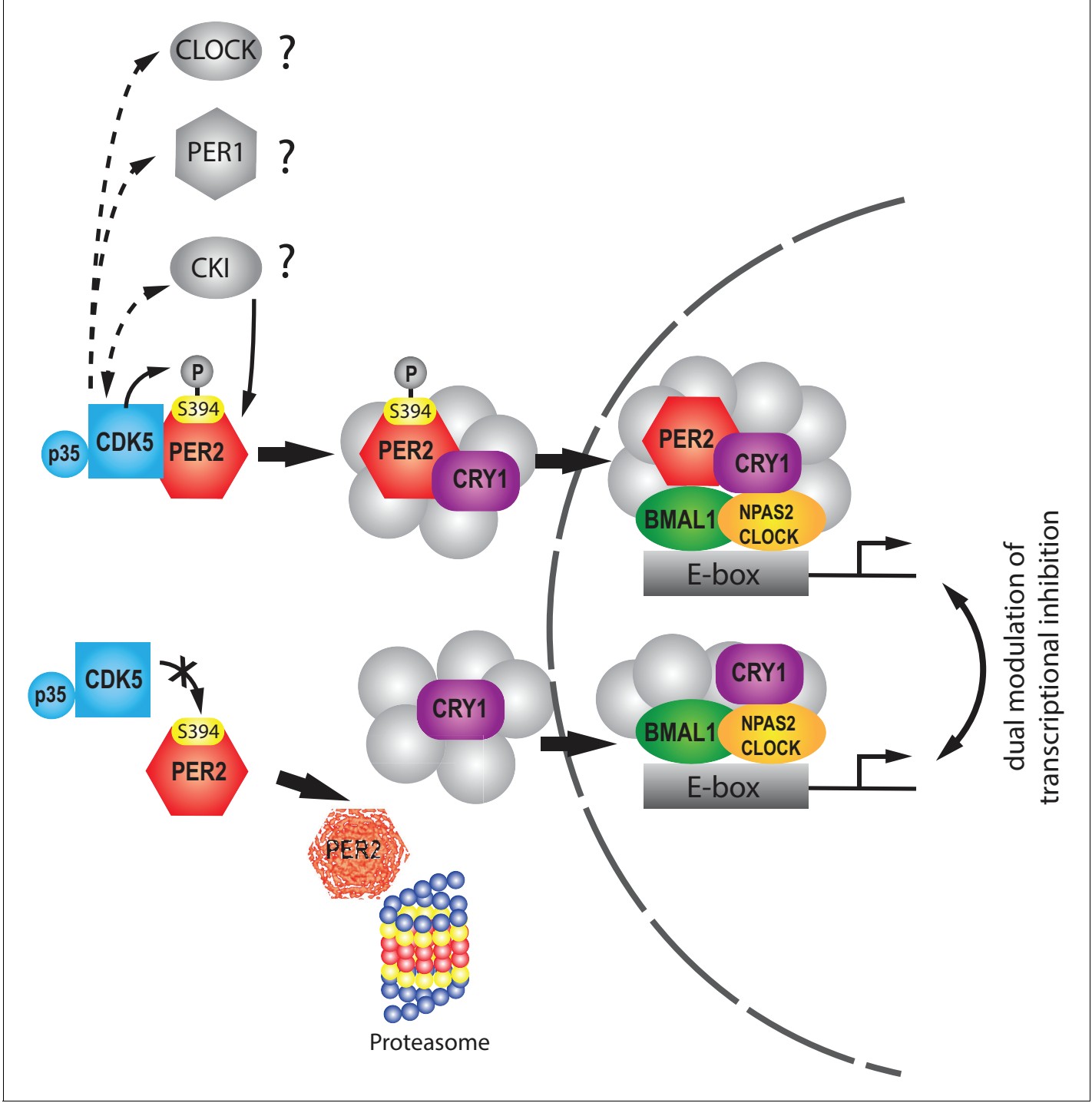

**Figure 7.** Model illustrating the regulation of PER2 by CDK5. The upper row illustrates phosphorylation of PER2 at S394 by CDK5 that subsequently favors interaction with CRY1 and leads to transport into the nucleus, where the PER2/CRY1 complex inhibits BMAL1/NPAS2 (or in the periphery CLOCK)-driven transcriptional activation. Of note is that CDK5 potentially phosphorylates other clock relevant components, such as CLOCK, PER1 and CKI. The lower part illustrates that inhibition of CDK5 leads to a lack of S394 PER2 phosphorylation, which renders the PER2 protein more prone to degradation by the proteasome. CRY1 does not form a complex with PER2 and hence PER2 is not transported into the nucleus. CRY1 enters the nucleus independently and can inhibit the BMAL1:NPAS2 (or in the periphery CLOCK) transcriptional complex. This model is consistent with the dual modulation of transcriptional inhibition (*Ye et al., 2014*; *Xu et al., 2015*). Transcriptional inhibition is modulated in an intricate unknown manner by various additional factors (gray) (*Aryal et al., 2017*) that may be cell type specific.

**Figure 6**. Pharmacological inhibition of CDK5 (**Figure 6A**), CRISPR/Cas9-mediated knock-out of *Cdk5* (**Figure 6B**), and shRNA-mediated knock-down of *Cdk5* (**Figure 6C**) all led to reduced levels of PER2 in cells. The half-life of PER2 is clearly increased in the presence of CDK5, rising from about 4 hr to 11 hr, indicating that phosphorylation at S394 has a stabilizing function. This is in accordance with previous results that described almost absent levels of PER2 in the *Per2^Brdm1* mutant mice (*Zheng et al., 1999*). This mouse strain expresses a PER2 lacking 87 amino acids in the PAS and PAC domains, where the S394 and the CDK5 consensus sequence are localized. CDK5 cannot phosphorylate this mutant PER2 and therefore the protein is less stable. As a consequence, the formation of the macromolecular complex responsible for nuclear transport of PER2 is disturbed. This results in a temporal change of BMAL1/CLOCK/NPAS2 activity, shortening the clock period. Accordingly, *Per2^Brdm1* mutant mice display a short period or no circadian period in constant darkness (*Zheng et al., 1999*), similar to the phenotype observed for the CDK5 knock-down mice (**Figure 2B**).

PER2 stability is affected by CK1δ/ε, which phosphorylate PER2 at several sites and regulate degradation of PER2 via the proteasome (*Eide et al., 2005*; *Xu et al., 2007*; *Narasimamurthy et al., 2018*). This effect is similar to the action of dbt on *Drosophila* per. Interestingly, CDK5 can phosphorylate CK1δ to reduce its activity (*Ianes et al., 2016*). This phosphorylation could cross-regulate the activities of both kinds of kinases to fine-tune the amount of PER2. This is evidenced by the observation, that knock-down of *Cdk5* in *Per2^Brdm1* mutant mice further shortens period in these animals (**Figure 2D,E**). The mammalian orthologue of shg, Gsk3β, does not phosphorylate the mammalian Tim but the nuclear receptor NR1D1 (*Mukherji et al., 2015*). This change in substrate may be related to the shift in function of the CRYs to replace Tim in the mammalian circadian oscillator. Similar to shg, CDK5 phosphorylation of PER2 increases its half-life (**Figure 6D**). Lack of CDK5, and therefore lack of phosphorylation at S394 of PER2, leads to proteasomal degradation of PER2 as evidenced by epoxomycin treatment, which inhibits the proteasome and reduces the decline of PER2 levels in the cell (**Figure 6E**). This is consistent with a recent report that describes the ubiquitin ligase MDM2 as controlling PER2 degradation via the proteasome (*Liu et al., 2018*). However, it is not clear whether it is the phosphorylation at S394 per se or the capacity to participate in a macromolecular complex to enter the nucleus that stabilizes PER2. In any case, this phosphorylation appears to be essential for nuclear entry of PER2 (**Figure 6F,G**).

A recent report showed that mammalian PER represses and de-represses transcription by displacing BMAL1-CLOCK from promoters in a CRY-dependent manner (*Chiou et al., 2016*). Our data support these findings. PER2 containing a S394G mutation, which abolishes CDK5-mediated phosphorylation, displayed reduced interaction potential with CRY1 (**Figure 6H**). Because CRY1 is involved in nuclear transport of PER2 (*Kume et al., 1999*; *Ollinger et al., 2014*; *Yagita et al., 2000*), lack of interaction with the S394G mutant form of PER2 leaves this protein in the cytoplasm, unable to enter the nucleus (**Figure 6G**). The present data are also in agreement with previous experiments in which we investigated the role of protein phosphatase 1 (PP1) and its effects on the circadian clock (*Schmutz et al., 2011*). Expression of a specific PP1 inhibitor in the brain lengthened circadian period and increased PER2 levels and its nuclear accumulation in neurons. These effects are all opposite to what we observe when PER2 is not phosphorylated at S394. Therefore, it could be speculated that PP1 is involved in the dephosphorylation of P-S394, thereby counterbalancing phosphorylation of this site by CDK5.

Taken together, our results indicate that CDK5 potentially affects several proteins that regulate circadian clock period. In particular, we find that CDK5 phosphorylates PER2 at S394. This phosphorylation appears to be important for PER2 to bind efficiently to CRY1 in order to allow entry of PER2 into the nucleus. Inhibition of CDK5 in cells leads to degradation of PER2 in the proteasome (**Figure 7**). Inhibition of CDK5 in vivo inhibits nuclear entry of PER2 and shortens period to a similar extent as observed in *Per2^Brdm1* mutant mice, which express a barely detectable level of protein lacking 87 amino acids including S394. Taken together, CDK5 regulates the circadian clock and influences PER2 nuclear transport via phosphorylation. Because PER2 is involved in several physiologically relevant pathways in addition to clock regulation (*Albrecht et al., 2007*), PER2 may mediate several biological functions that were previously linked to CDK5, such as the regulation of the brain reward system (*Benavides et al., 2007*; *Bibb et al., 2001*) and psychiatric diseases (*Engmann et al., 2011*; *Zhu et al., 2012*).

# Materials and methods

## Key resources table

| Reagent type (species) or resource | Designation | Source or reference | Identifiers | Additional information |
|---|---|---|---|---|
| Genetic reagent (*M. musculus*) | *Per2Brdm1* | Jackson Laboratory | Stock #: 003819 | PMID: 10408444 |
| Genetic reagent (*M. musculus*) | B6;129P2-Per2tm1Ual/Biat | European mouse mutant archive | Strain ID EM:10599 | PMID: 26838474 |
| Cell line (*M. musculus*) | NIH3T3 | ATCC | Cat. #: ATCCRCRL-1658Tm | Immortalized Mouse fibroblast cells |
| Cell line (*M. musculus*) | NIH3T3 CRISPR/Cas9 *Cdk5* KO | Origene | Cat. #: KN303042 | Immortalized Mouse fibroblast cells.7 ug/ml of puromycin are required for cells propagation |
| Cell line (*Human*) | HEK | ATCC | | Immortalized Kidney fibroblast cells |
| Transfected construct (*M. musculus*) | Sh RNA CDK5 plasmids | Origene | Cat. #: TL515615 A/B/C/D | |
| Transfected construct (*M. musculus*) | Sh RNA scramble | Origene | Cat. #: TR30021 | |
| Antibody | anti-PER2-1 (Rabbit polyclonal) | Alpha Diagnostic Lot # 869900A1.2-L | Cat. #: PER21-A RRID: AB_2236875: | 1:200 (IF) 1:50 (IP) 1:500/1:1000 (WB) |
| Antibody | anti-Cdk5 clone 2H6 (Mouse monoclonal) | Origene Lot # A001 | Cat. #: CF500397 RRID: AB_229166 | 1:20 (IF) 1:50 (IP) 1:500/1:1000 (WB) |
| Antibody | anti-GFP (Rabbit polyclonal) | Abcam | Cat. #: ab6556 RRID: AB_305564 | 1:500 (IF) |
| Antibody | anti-rabbit IgG (H+L) (Donkey polyclonal) | Alexa Fluor 488 Lot # 132876 | Cat. #: 711-545-152 RRID: AB_2313584 | 1:500 (IF) |
| Antibody | anti-mouse IgG (H+L) (Donkey polyclonal) | Alexa Fluor 647 Lot # 131725 | Cat. #: 715-605-150 RRID: AB_2340862 | 1:500 (IF) |
| Antibody | anti-rabbit IgG (H+L) (Donkey polyclonal) | Alexa Fluor 647 Lot # 136317 | Cat. #: 711-602-152 | 1:500 (IF) |
| Antibody | anti-HA (Mouse monoclonal) | Roche | Cat. #: 11583816001 RRID: AB_2532070 | 1:1000 (WB) |
| Antibody | anti-GST (Mouse monoclonal) | Sigma | Cat. #: G1160 RRID: AB_259845 | 1:1000 (WB) |
| Antibody | PER2 Phosphor Serine 133 (mouse monoclonal) | GenScript Company | Provided by the corresponding author | WB: 1:200 |
| Other | DAPI | Termofisher | Cat. #: D3571 RRID: AB_2307445 | (1 µg/mL) |
| Recombinant DNA reagent | Supplemental Table II | Complete list provided in the paper | | |
| Commercial assay or kit | pCR2.1-TOPO cloning | Thermofisher | Cat. #: K4500-01 | |
| Commercial assay or kit | QuikChange Site-Directed Mutagenesis Kit | Agilent | Cat. #: 200518 | |

*Continued on next page*

*Continued*

| Reagent type (species) or resource | Designation | Source or reference | Identifiers | Additional information |
|---|---|---|---|---|
| Chemical compound | Polyethylenimine, Linear, MW 25000, Transfection Grade (PEI 25K) | Polyscience Europe | Cat. #: 23966–1 | |
| Chemical compound | Roscovitine | Merk | Cat. #: R7772-1MG | |
| Chemical compound | Protein Agarose Beads | Roche | Cat. # 11 719 408 001 | |
| Chemical compound | cOmplete, EDTA-free Protease Inhibitor Cocktail | Merk | Cat. # 11873580001 | |
| Chemical compound | Isopropyl β-D-1-thiogalactopyranosid | Sigma-Aldrich | Cat. # 367-93-1 | |
| Chemical compound | L-Glutathione reduced | Merk | Cat. # 70-18-8 | |
| Chemical compound | Cycloheximide | Merk | Cat. # 66-81-9 | |
| Chemical compound | Epoxomicin | Sigma-Aldrich | Cat. # 134381-21-8 | |
| Peptide, recombinant protein | Cdk5/p35 Protein, active, 10 μg | Millipore | Cat. # 14–477 | |
| Peptide, recombinant protein | Histone H1 | Sigma-Aldrich | Cat. # H1917-100UG | |
| Software | Prism | GraphPad | Version 8.2.0 | |
| Software | ImageJ | ImageJ | Version 1.49 RRID: SCR_00370 | |
| Software | ClockLab | Actimetrics | Acquistion version: 3.208 Analysis version: 6.0.36 RRID: SCR_0114309 | |
| Software | Leica application Suite Advanced Fluorescence | Leica | Version 2.7.3.9723 | |

## Animals and housing

All mice were housed with food and water ad libidum in transparent plastic cages (267 mm long ×207 mm wide ×140 mm high; Techniplast Makrolon type 2 1264C001) with a stainless-steel wire lid (Techniplast 1264C116), kept in light- and soundproof ventilated chambers. All mice were entrained to a 12:12 hr LD cycle, and the time of day was expressed as Zeitgeber time (ZT; ZT0 lights on, ZT12 lights off). Two- to four-month-old males were used for the experiments. Housing as well as experimental procedures were performed in accordance with the guidelines of the Schweizer Tierschutzgesetz and the declaration of Helsinki. The state veterinarian of the Canton of Fribourg approved the protocol. The floxed Per2 mice (*Chavan et al., 2016*) are available at the European Mouse Mutant Archive (EMMA) strain ID EM:10599, B6;129P2-Per2[tm1Ual]/Biat.

## Synthetic dosage lethal (SDL) screen

The SDL screen was essentially performed as described earlier (*Measday et al., 2005*; *Tong, 2001*). Briefly, the bait strain Y2454 (MATα *mfa1Δ::MFA1pr-HIS3, can1Δ, his3Δ1, leu2Δ0, ura3Δ0, lys2Δ0*) carrying the plasmid YCplF2-*mPer2* (that drives expression of PER2 from the galactose-inducible *GAL1* promoter) was inoculated into 50 mL glucose-containing synthetic dropout medium lacking leucine (SD-Leu) and grown at 30°C overnight with shaking. Cells were then centrifuged, resuspended in 20 mL of the supernatant, poured into a sterile rectangular petri dish, spotted in a 96-well format on rectangular SD-Leu plates (coined 'bait plates' hereafter) using a Biomek 2000 robot (Beckman Coulter, USA), and then grown at 30°C for 3 days. In parallel, the gene deletion array in

the strain BY4741 (MATa *his3Δ1, leu2Δ0, met15Δ0, ura3Δ0*) was spotted from the storage plates onto fresh G418-containing YPD plates (96-well format) and also grown at 30°C for 3 days. For the mating procedure (overnight at 30°C), colonies from bait plates were (robotically) spotted onto plates containing YPD (plus adenine) and the colonies from the gene deletion array plates were (each separately and in duplicate) spotted on top of them. The next day, the colonies were transferred to G418-containing SD plates lacking lysine, methionine, and leucine (SD-Lys/Met/Leu/+G418) to select for diploids that harbour the YCplF2-*mPer2* plasmid. Diploids were then spotted onto plates containing sporulation medium (10 g L$^{-1}$ potassium acetate, 1 g L$^{-1}$ yeast extract, 0.1 g L$^{-1}$ glucose, 2% w/v agar, supplemented with uracil, histidine, and G418) and incubated at 24°C. After 9 days, tetrads were observed and the colonies were transferred to canavanine-containing SD plates lacking arginine and histidine (SD-Arg/His/+canavanine) to select for MATa haploids. Following growth at 30°C for three days, a second haploid selection was carried out by spotting the colonies on SD-Arg/His/Leu/+canavanine plates (to select for MATa haploids containing the YCplF2-*mPer2* plasmid). Following growth at 30°C for 2 days, a third haploid selection was carried out by spotting cells on SD-Arg/His/Leu/+canavanine/+G418 plates (to select for MATa haploids containing the YCplF2-*mPer2* plasmid as well as the respective gene deletions of the yeast knockout collection). Following incubation at 30°C for 5 days, colonies were then spotted in parallel onto SD-Arg/His/Leu/+G418 plates and on SD-Raf/Gal-Arg/His/Leu/+G418 plates (containing 1% raffinose and 2% galactose as carbon sources) to induce expression of PER2. Both types of plates were incubated at 30°C for 4 days and photographed every day. Strains that grew significantly less well on SD-Raf/Gal-Arg/His/Leu/+G418 than on SD-Arg/His/Leu/+G418 included *eap1Δ*, *gnd1Δ*, and *pho85Δ*. In control experiments, the respective original yeast knockout collection mutants were transformed in parallel with the YCplF2-*mPer2* or the empty YCplF2 plasmid (*Foreman and Davis, 1994*), selected on SD-Leu plates, grown overnight in liquid SD-Leu, spotted (10-fold serial dilutions) on SD-Raf/Gal-Leu plates, and grown for 3 days at 30° (*Figure 1A*). Please note that all media containing G418 were made with glutamate (1 g L$^{-1}$) instead of ammonium sulfate as nitrogen source, as recommended in *Tong (2001)*.

## Adeno Associate Virus (AAV) production and stereotaxic injections

Adeno Associate Viruses (AAVs) were produced in the Viral Vector Facility (ETH Zurich). Plasmids used for the production are available on the VVF web site. Two constructs were produced. ssAAV-9/2-hSyn1-chI[mouse(shCdk5)]-EGFP-WPRE-SV40p(A) carried the shRNA against Cdk5 (shD, see *Figure 2—figure supplement 1* and *Supplementary file 2*) which knocked down only neuronal *Cdk5*. ssAAV-9/2-hSyn1-chI[1x(shNS)]-EGFP-WPRE-SV40p(A) was the scrambled control.

Stereotaxic injections were performed on 8-week-old mice under isofluorene anaesthesia using a stereotaxic apparatus (Stoelting). The brain was exposed by craniotomy and the Bregma was used as reference point for all coordinates. AAVs were injected bilaterally into the SCN (Bregma: anterior-posterior (AP) − 0.40 mm; medial-lateral (ML) ±0.00 mm; dorsal-ventral (DV) – 5.5 mm, angle + /- 3°) using a hydraulic manipulator (Narishige: MO-10 one-axis oil hydraulic micromanipulator, http://products.narishige-group.com/group1/MO-10/electro/english.html) at a rate of 40 nL/min through a pulled glass pipette (Drummond, 10 µl glass micropipet; Cat number: 5-000-1001-X10). The pipette was first raised 0.1 mm to allow spread of the AAVs, and later withdrawn 5 min after the end of the injection. After surgery, mice were allowed to recover for 2 weeks and entrained to LD 12:12 prior to behavior and molecular investigations.

## Locomotor activity monitoring

Analysis of locomotor activity parameters was done by monitoring wheel-running activity, as described in *Jud et al. (2005)*, and calculated using the ClockLab software (Actimetrics). Briefly, for the analysis of free-running rhythms, animals were entrained to LD 12:12 and subsequently released into constant darkness (DD). Internal period length (τ) was determined from a regression line drawn through the activity onsets of ten days of stable rhythmicity under constant conditions. Total and daytime activity, as well as activity distribution profiles, was calculated using the respective inbuilt functions of the ClockLab software (Acquisition Version 3.208, Analysis version 6.0.36). Numbers of animals used in the behavioral studies are indicated in the corresponding figure legends.

## Immunofluorescence

Animals used for the immunohistochemistry were killed at appropriate ZTs indicated in the corresponding figure legends. Brains were perfused with 0.9% NaCl and 4% PFA. Perfused brains were cryoprotected by 30% sucrose solution and sectioned (40 μm, coronal) using a cryostat. Sections chosen for staining were placed in 24-well plates (two sections per well), washed three times with 1x TBS (0.1 M Tris/0.15 M NaCl) and 2x SSC (0.3 M NaCl/0.03 M tri-Na-citrate pH 7). Antigen retrieval was performed with 50% formamide/2x SSC by heating to 65°C for 50 min. Then, sections were washed twice in 2x SSC and three times in 1x TBS pH 7.5, before blocking them for 1.5 hr in 10% fetal bovine serum (Gibco)/0.1% Triton X-100/1x TBS at RT. After the blocking, the primary antibodies, rabbit anti-PER2-1 1:200 (Alpha Diagnostic, Lot numb. 869900A1.2-L), mouse anti-Cdk5 clone 2H6 1:20 (Origene, Lot numb. A001), and rabbit anti-GFP 1:500 (abcam ab6556) diluted in 1% FBS/ 0.1% Triton X-100/1x TBS, were added to the sections and incubated overnight at 4°C. The next day, sections were washed with 1x TBS and incubated with the appropriate fluorescent secondary antibodies diluted 1:500 in 1% FBS/0.1% Triton X-100/1x TBS for 3 hr at RT. (Alexa Fluor 488-AffiniPure Donkey Anti-Rabbit IgG (H+L) no. 711–545–152, Lot: 132876, Alexa Fluor647-AffiniPure Donkey Anti-Mouse IgG (H+L) no. 715–605–150, Lot: 131725, Alexa Fluor647-AffiniPure Donkey Anti-Rabbit IgG (H+L) no. 711–602–152, Lot: 136317 and all from Jackson Immuno Research). Tissue sections were stained with DAPI (1:5000 in PBS; Roche) for 15 min. Finally, the tissue sections were washed again twice in 1x TBS and mounted on glass microscope slides. Fluorescent images were taken by using a confocal microscope (Leica TCS SP5), and images were taken with a magnification of 40x or 63x. Images were processed with the Leica Application Suite Advanced Fluorescence 2.7.3.9723 according to the study by *Schnell et al. (2014)*.

Immunostained sections were quantified using ImageJ version 1.49. Background was subtracted and the detected signal was divided by the area of measurement. An average value obtained from three independent areas for every section was used. The signal coming from sections obtained from silenced mice was quantified as relative amount to the scramble, which was set to 1. Statistical analysis was performed on three animals per treatment.

## Cell culture

NIH3T3 mouse fibroblast cells (ATCCRCRL-1658) were maintained in Dulbecco's modified Eagle's medium (DMEM), containing 10% fetal calf serum (FCS) and 100 U/mL penicillin-streptomycin at 37° C in a humidified atmosphere containing 5% $CO_2$. Cdk5 KO cells were produced applying the CRISPR/Cas9 technique according to the manufacturer's protocol of the company (Origene, SKU # KN303042).

## Plasmids

The following plasmids used were previously described: pSCT-1, pSCT-1mPer2, pSCT-1 mPer-V5, pSCT1 ΔPasA mPer2 -V5, pSCT1 ΔPasB mPer2 -V5 (*Langmesser et al., 2008*) (*Schmutz et al., 2010*). pSCT-1 Cdk5-HA, pet-15b Cdk5-HIS, Gex-4T Per2 1–576, pGex-4T Per2 577–1256 were produced for this paper. The full-length cDNA (or partial fragments) encoding PER2 and the full-length Cdk5 were previously sub-cloned in the TOPO vector according to the manufacturer's protocol (Catalog numbers pCR2.1-TOPO vector: K4500-01). They were subsequently transferred into the plasmid pSCT-1 using appropriate restriction sites. pGex-4T Per2 1–576 S394G, S394D, pSCT-1 mPer2 S394G were obtained using site-specific mutagenesis according to the manufacturer's protocol on the requested codon carrying the interested amino acid of interest (Agilent Catalog # 200518). For accession numbers, vectors, mutations, and primers sources, see *Supplementary file 2*.

## Transfection and co-immunoprecipitation of overexpressed proteins

NIH 3T3 cells were transfected in 10 cm dishes at about 70% of their total confluency using linear polyethylenimine (LINPEI25; Polysciences Europe). The amounts of expression vectors were adjusted to yield comparable levels of expressed protein. Thirty hours after transfection, protein extracts were prepared. With regard to immunoprecipitation, each antibody mentioned in the paper was used in the conditions indicated by the respective manufacturer. The next day, samples were captured with 50 μL at 50% (w/v) of protein-A agarose beads (Roche) at 50% (w/v) and the reaction was kept at 4°C for 3 hr on a rotary shaker. Prior to use, beads were washed three times with the

appropriate protein buffer and resuspended in the same buffer (50% w/v). The beads were collected by centrifugation and washed three times with NP-40 buffer (100 mM Tris-HCl pH7.5, 150 mM NaCl, 2 mM EDTA, 0.1% NP-40). After the final wash, beads were resuspendend in 2% SDS, 10% glycerol, 63 mM Trish-HCL pH 6.8 and proteins were eluted for 15 min at RT. Laemmli buffer was finally added, samples were boiled for 5 min at 95° C and finally loaded onto 10% SDS-PAGE gels (*Laemmli, 1970*).

## Total protein extraction from cells (Ripa method)
Medium was aspirated from cell plates, which were washed two times with 1x PBS (137 mM NaCl, 7.97 mM $Na_2HPO_4 \times 12\ H_2O$, 2.68 mM KCl, 1.47 mM $KH_2PO_4$). Then PBS was added again and plates were kept 5 min at 37°C. NHI3T3 or HEK cells were detached and collected in tubes and washed two times with 1x PBS. After the last washing, pellets were frozen in liquid $N_2$, resuspended in Ripa buffer (50 mM Tris-HCl pH7.4, 1% NP-40, 0.5% Na-deoxycholate, 0.1% SDS, 150 mM NaCl, 2 mM EDTA, 50 mM NaF) with freshly added protease or phosphatase inhibitors, and homogenized by using a pellet pestle. After that samples were centrifuged for 15 min at 16,100 g at 4°C. The supernatant was collected in new tubes and pellet discarded.

## Total protein extraction from brain tissue
Total brain or isolated SCN tissue was frozen in liquid $N_2$, and resuspended in lysis buffer (50 mM Tris-HCl pH 7.4, 150 mM NaCl, 0.25% SDS, 0.25% sodium deoxycholate, 1 mM EDTA) and homogenized by using a pellet pestle. Subsequently, samples were kept on ice for 30 min and vortexed five times for 30 s each time. The samples were centrifuged for 20 min at 12,000 rpm at 4°C. The supernatant was collected in new tubes and the pellet discarded.

## Nuclear/cytoplasmic fractionation
Tissues or cells were resuspended in 100 mM Tris-HCl pH 8.8/10 mM DTT and homogenized with a disposable pellet pestle. After 10 min incubation on ice, the samples were centrifuged at 2500 g for 2 min at 4°C and the supernatant discarded. After adding 90 µL of completed cytoplasmic lysis buffer (10 mM EDTA, 1 mM EGTA, 10 mM Hepes pH 6.8, 0.2% Triton X-100, protease inhibitor cocktail (Roche), NaF, PMSF, ß-glycerophosphate), the pellet was resuspended by vortexing, followed by centrifugation at 5200 rpm for 2 min at 4°C. The supernatant transferred into a fresh 1.5 mL tube was the CYTOPLASMIC EXTRACT. The pellet was washed three times with cytoplasmic lysis buffer and resuspended in 45 µL 1x NDB (20% glycerol, 20 mM Hepes pH 7.6, 0.2 mM EDTA, 2 mM DTT) containing 2x proteinase and phosphatase inhibitors followed by adding 1 vol of 2x NUN (2 M Urea, 600 mM NaCl, 2% NP-40, 50 mM Hepes pH 7.6). After vortexing the samples were incubated 30 min on ice, centrifuged 30 min at 13,000 rpm at 4°C and the supernatant that was transferred into a fresh tube was the NUCLEAR EXTRACT.

## Immunoprecipitation using brain tissue extracts
A protein amount corresponding to between 400 and 800 µg of total extract was diluted with the appropriate protein lysis buffer in a final volume of 250 µL and immunoprecipitated using the indicated antibody (ratio 1:50) and the reaction was kept at 4°C overnight on a rotary shaker. The day after, samples were captured with 50 µL of 50% (w/v) protein-A agarose beads (Roche) and the reaction was kept at 4°C for 3 hr on a rotary shaker. Prior to use, beads were washed three times with the appropriate protein buffer and resuspended in the same buffer (50% w/v). The beads were collected by centrifugation and washed three times with NP-40 buffer (100 mM Tris-HCl pH7.5, 150 mM NaCl, 2 mM EDTA, 0.1% NP-40). After the final wash, beads were resuspendend in 2% SDS 10%, glycerol, 63 mM Trish-HCL pH 6.8 and proteins were eluted for 15 min at RT. Laemmli buffer was finally added, samples were boiled 5 min at 95° C and loaded onto 10% SDS-PAGE gels.

## Pull-down assay with GST-Per2 fragments
GST-fused recombinant Per2 proteins were expressed in the *E. coli* Rosetta strain [plasmids: GST-Per2 N-M (1-576), GST-Per2 M-C (577-1256)]. Proteins were induced with 1 mM IPTG (Sigma-Aldrich) for 3 hr at 30°C. Subsequently, proteins were extracted in an appropriate GST lysis buffer (50 mM Tris-Cl pH 7.5, 150 mM NaCl, 5% glycerol) adjusted to 0.1% Triton X-100 and purified to

homogeneity with glutathione-agarose beads for 2 hr at 4°C. The beads were then incubated overnight at 4°C and washed with GST lysis buffer adjusted with 1 mM DTT. Subsequently, elution with 10 mM reduced glutathione took place for 15 min at room temperature. Elution was stopped by adding Laemmli buffer and samples were loaded onto the gel after 5 min at 95°C and WB was performed using anti-GST (Sigma no. 06–332) and anti-HA antibodies (Roche no. 11867423001) for immunoblotting.

## CRISPR/Cas9 *Cdk5* knock-out cell line

The CRISPR/Cas9 Cdk5 cell line was produced starting from NIH3T3 cells using a kit provided by Origene (www.origene.com). The knock-out cell line was produced according to the manufacturer's protocol. Briefly, cells at 80% of confluency were co-transfected with a donor vector containing the homologous arms and functional cassette, and the guide vector containing the sequence that targets the *Cdk5* gene. In parallel, a scrambled negative guide was also co-transfected with a donor vector. 48 hr after transfection the cells were split 1:10 and grown for 3 days. Cells were split another seven times (this time is necessary to eliminate the episomal form of donor vector, in order to have only integrated forms). Then, single colonies were produced and clones were analyzed by PCR in order to find those clones that did not express *Cdk5* RNA. Positive clones were re-amplified.

PCR primers for genomic Cdk5:
FW: 5'-tgtgagtaccacctcctctgcaa-3'
RW: 5'-ttaaacaggccaggcccc-3'

## Knockdown of Cdk5

About 24 hr after seeding cells, different shRNA Cdk5 plasmids (Origene TL515615 A/B/C/D Cdk5 shRNA) were transfected to knock down *Cdk5* according to the manufacturer's instructions. The knock-down efficiency was assessed at 48 hr post transduction by western blotting. Scrambled shRNA plasmid (Origene TR30021) was used as a negative control.

## Cycloheximide treatment

NIH3T3 cells were treated with 100 µM cycloheximide 48 hr after transfection with the indicated vectors, and cells were harvested 0, 3, and 6 hr after treatment.

## Proteasome inhibitor treatment

About 48 hr after transfection with either scrambled or shCdk5, cells where *Cdk5* was silenced were treated for 12 hr with either DMSO (vehicle) or epoxomicin (Sigma-Aldrich) at a final concentration of 0.2 µM. Samples were collected, and proteins extracted followed by western blotting.

## In vitro kinase assay

Recombinant GST-fused PER2 protein fragments were expressed and purified from the BL21 Rosetta strain of *E. coli* according to the manufacturer's protocol described before, using glutathione-sepharose affinity chromatography (GE Healthcare). Each purified protein (1 µg) was incubated in the presence or absence of recombinant Cdk5/p35 (the purified recombinant N-terminal His6-tagged human Cdk5 and N-terminal GST-tagged human p25 were purchased from Millipore). Reactions were carried out in a reaction buffer (30 mM Hepes, pH 7.2, 10 mM MgCl2, and 1 mM DTT) containing [γ-$^{32}$P] ATP (10 Ci) at room temperature for 1 hr and then terminated by adding SDS sample buffer and boiling for 10 min. Samples were subjected to SDS-PAGE, stained by Coomassie Brilliant Blue, and dried, and then phosphorylated proteins were detected by autoradiography.

## In vitro kinase assay using immunoprecipitated Cdk5 from SCN

CDK5 was immunoprecipitated from SCN samples at different ZTs (circa 600 µg of protein extract) (*Figure 8*). After immunoprecipitation at 4°C overnight with 2x Protein A agarose (Sigma-Aldrich), samples were diluted in washing buffer and split in two halves. One half of the IP was used for an in vitro kinase assay. Briefly, 1 µg of histone H1 (Sigma-Aldrich) was added to the immunoprecipitated CDK5 and assays were carried out in reaction buffer (30 mM Hepes, pH 7.2, 10 mM MgCl$_2$, and 1 mM DTT) containing [γ-$^{32}$P] ATP (10 Ci) at room temperature for 1 hr and then terminated by adding SDS sample buffer and boiling for 5 min. Samples were subjected to 15% SDS-PAGE, stained by

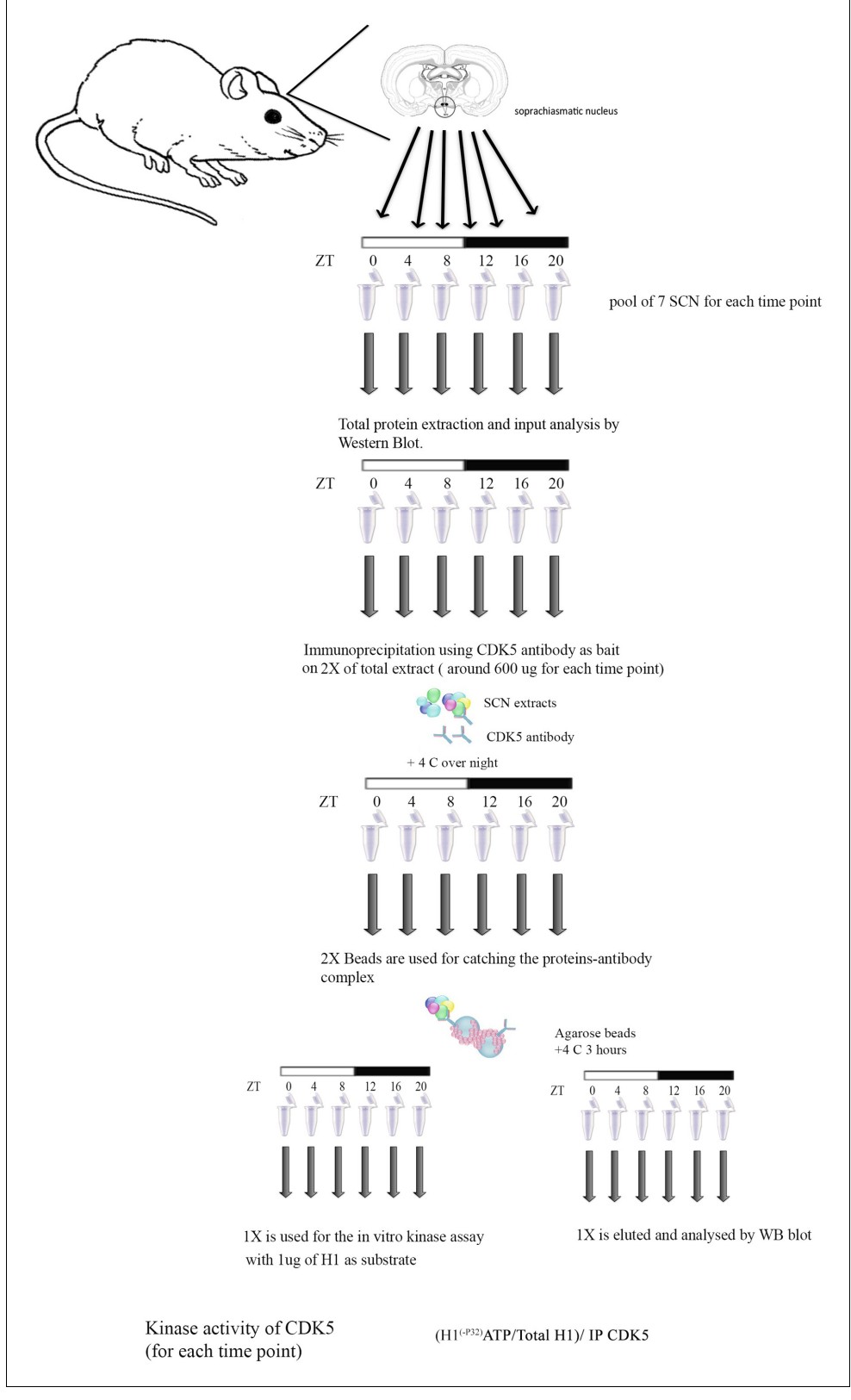

**Figure 8.** Workflow of the in vitro kinase assay. Workflow of the in vitro kinase assay performed using immunoprecipitated CDK5 from SCN protein extracts is schematized here. Seven mice were sacrificed, SCN tissues were isolated and pooled together every 4 hr starting from ZT 0 (lights on) until ZT20 (ZT12 lights off). Total protein was obtained from each pool of tissues, the quality of the extracts was checked by WB, and subsequently CDK5 was immunoprecipitated at each time point. Agarose beads detained the immunoprecipitation and one half of the precipitate was used for an in vitro

*Figure 8 continued on next page*

*Figure 8 continued*

kinase assay using as substrate commercial histone H1 as substrate. The other half was analyzed by WB in order to quantify the amount of protein immunoprecipitated, which was used for the kinase assay. Kinase activity around the clock was quantified using the following formula: ($^{32}$P-H1/total H1)/ amount of immunoprecipitated CDK5.

Coomassie Brilliant Blue, and dried, and then phosphorylated histone H1 was detected by autoradiography. The other half of the IP was used for Western blotting to determine the total amount of CDK5 immunoprecipitated from the SCN samples. To quantify the kinase activity at each time point, the following formula was used: ([$^{32}$P] H1/total H1 for each reaction)/CDK5 IP protein.

## Filter-aided in vitro kinase assay, phosphopeptide enrichment and mass spectrometry analyses

Filter-aided in vitro kinase assays and mass spectrometry analyses were performed essentially as described (*Hatakeyama et al., 2019*). Briefly, recombinant Cdk5/p35 (Millipore) was incubated with the GST-fused PER2 protein fragment. On 10 kDa MW-cutoff filters (PALL) samples were incubated in kinase buffer containing 50 mM Hepes, pH 7.4, 150 mM NaCl, 0.625 mM DTT, Phostop tablets (Roche), 6.25 mM MgCl$_2$, and 1.8 mM ATP at 30°C for 1 hr. Samples without ATP were used as negative control. Assays were quenched by 8 M urea and 1 mM DTT. Protein digestion for MS analysis was performed overnight (*Wiśniewski et al., 2009*). Phosphopeptides were enriched by metal oxide affinity enrichment using titanium dioxide (GL Sciences Inc, Tokyo, Japan) (*Zarei et al., 2016*).

LC-MS/MS measurements were performed on a QExactive Plus mass spectrometer coupled to an EasyLC 1000 nanoflow-HPLC. Peptides were separated on fused silica HPLC-column tip (I.D. 75 µm, New Objective, self-packed with ReproSil-Pur 120 C18-AQ, 1.9 µm [Dr. Maisch, Ammerbuch, Germany] to a length of 20 cm) using a gradient of A (0.1% formic acid in water) and B (0.1% formic acid in 80% acetonitrile in water): loading of sample with 0% B with a flow rate of 600 nL min-1; separation ramp from 5–30% B within 85 min with a flow rate of 250 nL min-1. NanoESI spray voltage was set to 2.3 kV and ion-transfer tube temperature to 250°C; no sheath and auxiliary gas was used. Mass spectrometers were operated in the data-dependent mode; after each MS scan (mass range m/z = 370–1750; resolution: 70,000) a maximum of 10 MS/MS scans were performed using a normalized collision energy of 25%, a target value of 1000 and a resolution of 17,500. The MS raw files were analyzed using MaxQuant Software version 1.4.1.2 (*Cox and Mann, 2008*) for peak detection, quantification and peptide identification using a full-length Uniprot Mouse database (April, 2016) and common contaminants such as keratins and enzymes used for digestion as reference. Carbamidomethylcysteine was set as fixed modification and protein amino-terminal acetylation, serine-, threonine- and tyrosine-phosphorylation, and oxidation of methionine were set as variable modifications. The MS/MS tolerance was set to 20 ppm and three missed cleavages were allowed using trypsin/P as enzyme specificity. Peptide, site and protein FDR based on a forwards-reverse database were set to 0.01, minimum peptide length was set to 7, and minimum number of peptides for identification of proteins was set to one, which must be unique. The 'match-between-run' option was used with a time window of 1 min. The mass spectrometry proteomics data have been deposited to the ProteomeXchange Consortium via the PRIDE partner repository with the dataset identifier PXD012068 (project name: Cyclin dependent kinase 5 (CDK5) regulates the circadian clock; project accession: PXD012068).

## Generation of an antibody against phospho-serine 394

We raised in mouse a specific monoclonal antibody recognizing the phosphorylated form of serine 394 of PER2 in collaboration with GenScript Company. The sequence used for the immunogen preparation was: FDY {pSer} PIRFRTRNGEC. 3 Balb/c mice and 3 C57 mice were immunized with conventional strategies and antisera obtained from those animals were used for the first control experiment performed by in vitro kinase assay (*Figure 5—figure supplement 3*). The positive antiserum was used for the cell fusions. Subsequently, a screening with 16 96-well plates (from 2 × 10E4 clones) was performed by indirect ELISA, primary screening with phospho-peptide, then counter-screening with non-phospho-peptide. The obtained supernatants were tested by in vitro kinase assay in order to screen which one was better recognized the phospho-form of PER2 S394 (*Figure 5—figure*

*supplement 4*). Finally, five selected positive primary clones selected were subcloned by limiting dilution and tested as final antibody (*Figure 5—figure supplement 5*).

## Statistical analysis

Statistical analysis of all experiments was performed using GraphPad Prism6 software. Depending on the type of data, either an unpaired t-test, or one- or two-way ANOVA with Bonferroni or Tukey's post-hoc test was performed. Values considered significantly different are highlighted. [$p < 0.05$ (*), $p < 0.01$ (**), or $p < 0.001$ (***)].

# Acknowledgements

We thank Stéphanie Aebischer, Antoinette Hayoz, Cressida Harvey, Naila Ben Fredi, Jean-Charles Paterna (Viral Vector Facility, University of Zürich) and the Bioimage platform (University of Fribourg) for technical support. Funding from the Swiss National Science Foundation (31003A_166682, 310030_166474/1 and 316030_177088) is acknowledged. AB was supported by a fellowship from the Fondazione Cenci Bolognetti, Instituto Pasteur.

# Additional information

### Competing interests

Elisabetta Cameroni: Elisabetta Cameroni is affiliated with Humabs Biomed SA. The author has no other competing interests to declare. The other authors declare that no competing interests exist.

### Funding

| Funder | Grant reference number | Author |
| --- | --- | --- |
| Fondazione Cenci Bolognetti, Instituto Pasteur | | Andrea Brenna |
| Schweizerischer Nationalfonds zur Förderung der Wissenschaftlichen Forschung | 31003A_166682 | Urs Albrecht |
| Schweizerischer Nationalfonds zur Förderung der Wissenschaftlichen Forschung | 310030_166474/1 | Claudio De Virgilio |
| Schweizerischer Nationalfonds zur Förderung der Wissenschaftlichen Forschung | 316030_177088 | Jörn Dengjel |

The funders had no role in study design, data collection and interpretation, or the decision to submit the work for publication.

### Author contributions

Andrea Brenna, Conceptualization, Data curation, Formal analysis, Validation, Investigation, Methodology, Writing—review and editing; Iwona Olejniczak, Rohit Chavan, Investigation, Methodology; Jürgen A Ripperger, Data curation, Formal analysis, Supervision, Investigation, Methodology, Writing—review and editing; Sonja Langmesser, Formal analysis, Investigation, Methodology, Writing—review and editing; Elisabetta Cameroni, Zehan Hu, Formal analysis, Investigation; Claudio De Virgilio, Jörn Dengjel, Resources, Data curation, Methodology, Writing—review and editing; Urs Albrecht, Conceptualization, Resources, Data curation, Supervision, Funding acquisition, Visualization, Writing—original draft, Project administration

### Author ORCIDs

Andrea Brenna http://orcid.org/0000-0002-8542-9855
Jürgen A Ripperger http://orcid.org/0000-0002-9345-5172
Elisabetta Cameroni https://orcid.org/0000-0002-9102-8943

Claudio De Virgilio [ID] http://orcid.org/0000-0001-8826-4323
Jörn Dengjel [ID] http://orcid.org/0000-0002-9453-4614
Urs Albrecht [ID] https://orcid.org/0000-0002-0663-8676

## Ethics

Animal experimentation: This study was performed in strict accordance with the recommendations in the Guide for the Care and Use of Laboratory Animals of the Swiss Legislation and the declaration of Helsinki. The protocols were approved by the state veterinarian of the State of Fribourg (Permit Number: 2015-33).

## Decision letter and Author response

Decision letter https://doi.org/10.7554/eLife.50925.sa1
Author response https://doi.org/10.7554/eLife.50925.sa2

## Additional files

### Supplementary files

• Supplementary file 1. Phosphorylation sites of GST-Per2 (1-576) detected by mass spectrometry. The serine at position 394 stands out as the best localized phosphorylation site within a CDK5 consensus motif with a high peptide score (highlighted in yellow). The colored diagram shows the structural elements of PER2 (1–576) with the S394 phosphorylation site indicated. PEP: posterior error probability; Loc. Prob.; localization probability.

• Supplementary file 2. Plasmids.

• Transparent reporting form

### Data availability

Data supporting the findings of this work are available within the paper and its supplementary files, and on Dryad (https://doi.org/10.5061/dryad.4067r78). Non-commercial biological materials are provided upon request to the corresponding author. Proteomics data have been deposited to the ProteomeXchange Consortium via the PRIDE partner repository with the dataset identifier PXD012068. The Per2Brdm1 mutant mouse strain is available at the Jackson Laboratory Stock No: 003819 (B6. Cg-Per2 tm1Brd Tyr c-Brd). The floxed Per2 mice are available at the European Mouse Mutant Archive (EMMA) strain ID EM:10599, B6;129P2-Per2tm1Ual/Biat.

The following datasets were generated:

| Author(s) | Year | Dataset title | Dataset URL | Database and Identifier |
|---|---|---|---|---|
| Brenna A, Olejnic-zak I, Chavan R, Ripperger J, Langmesser S, Cameroni E, Hu Z, Virgilio CD, Dengjel J, Albrecht U | 2019 | Data from: Cyclin dependent kinase 5 (CDK5) regulates the circadian clock | https://doi.org/10.5061/dryad.4067r78 | Dryad Digital Repository, 10.5061/dryad.j1fd7 |
| Brenna A, Olejnic-zak I, Chavan R, Ripperger J, Langmesser S, Cameroni E, Hu Z, Virgilio CD, Dengjel J, Albrecht U | 2019 | Cyclin dependent kinase 5 (CDK5) regulates the circadian clock | http://proteomecentral. proteomexchange.org/ cgi/GetDataset?ID= PXD012068 | ProteomeXchange, PXD012068 |

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
