## [Decision Letter]

**Acceptance summary:**

It is clear that post-translational modifications of PER2 are critically important to regulating circadian timing. There are a growing number of modifying enzymes and different type of PTM that form a complex network of regulation. The addition of CDK5 to this network is an important one.

**Decision letter after peer review:**

Thank you for submitting your article "Cyclin dependent kinase 5 (CDK5) regulates the circadian clock" for consideration by *eLife*. Your article has been reviewed by two peer reviewers, and the evaluation has been overseen by a Reviewing Editor and Catherine Dulac as the Senior Editor. The reviewers have opted to remain anonymous.

The reviewers have discussed the reviews with one another and the Reviewing Editor has drafted this decision to help you prepare a revised submission.

Summary:

Brenna and colleagues provide in this article interesting evidences of the role of CDK5 in the regulation of the circadian clock. Using genetic screen in yeast, viruses-injected animals and knockout and knockdown cells, they clearly show CDK5 shows a rhythmic activity in the SCN, interact with PER2 in a rhythmic fashion, phosphorylates PER2 at serine 394 and regulates its stability and nuclear transport. Finally, they show that Cdk5 knockdown in the SCN of injected animals reduces their circadian period. While authors convincingly show that CDK5 regulates PER2 stability and transport through phosphorylation, the impact of this regulation on the reduced period observed in Cdk5 knockdown animals is less clear. This is confirmed by the fact that this knockdown has the same effect on the period of Per2 mutant animals, suggesting alternative targets that need more attention.

However, the contribution of the described pathway to the observed changes of circadian period length upon knockdown of CDK5 in the SCN is not entirely convincing and is based on circumstantial evidence. Specifically, CDK5 could act on CK1δ, as described by the Virshup lab. CK1α which is not phosphorylated at T347 is more active, resulting in a short period phenotype, just as the one observed by the authors.

Essential revisions:

- Figure 1. The short period phenotype in CDK5 knockdown animals appears to be Per2-independent since knockdown of CDK5 in *Per2^Brdm1^* animals still shows comparable period reduction as knockdown of CDK5 in wild type mice. *Per2^Brdm1^* still express a truncated form of Per2, hence the authors should also consider a similar experiment in Per2 KO animals to prove Per2-dependent mechanism. Alternatively, the CDK5 phenotype in *Per2^Brdm1^*animals could be due to CDK5 phosphorylating Per1, which is not discussed by the authors.

- Figure 1. Eng, Edison and Virshup, 2017, demonstrated that CKIδ activity towards Per2 is regulated via phosphorylation of T347 by proline-directed kinases (presumably also CDK5). This possibility should be addressed by the authors.

- Figure 6. To prove the relevance of this mechanism to other peripheral oscillators, the authors should include an experiment in NIH-3T3 Bmal1-luc cells transfected with shRNA against CDK5.

- Mutation of PER2 Ser 394 impacts circadian oscillation.

- CDK5 potentially also phosphorylates and regulates PER1 stability and transport.

- CDK5 regulation of CLOCK (and potentially NPAS2) contributes to the observed phenotype: CLOCK stability and transport are impacted by Cdk5 knockout? Is the circadian effect of Cdk5 knockdown also observed in Clock knockout?

While the effect of CDK5 on circadian period is clearly showed, the involvement of PER2 in the mechanism is not clear, meaning the model on Figure 7 is mainly speculative at this stage.

---

## [Author Response]

Essential revisions:- Figure 1. The short period phenotype in CDK5 knockdown animals appears to be Per2-independent since knockdown of CDK5 in Per2^Brdm1^ animals still shows comparable period reduction as knockdown of CDK5 in wild type mice.

We think the reviewer is referring to Figure 2 and not Figure 1. Figure 2 shows that knock-down of CDK5 causes short period length that is comparable to the period of *Per2^Brdm1^*mutant mice. As the reviewer states correctly, knockdown of Cdk5 in *Per2^Brdm1^* mutant mice leads to even further shortening of period. The reviewer concludes from this that period shortening by knockdown of Cdk5 is therefore Per2 independent.

A genetic complementation assay, in our case the knockdown of Cdk5 in *Per2^Brdm1^*, allows the identification of genes that are in the same complementation group. If Per2 was the sole target of CDK5 and the phosphorylation important for the period length, then one would expect that the combination of knockdown and knockout would have the same effect as the single conditions. This is not what was observed. However, the effect is not simply additive to indicate two different pathways as suggested by the reviewer. In this scenario, the one-hour reduction of *Per2^Brdm1^* and the one-hour reduction of the knockdown would add up to an at least two-hour shortening and not to the about 1.5 hour shortening as observed. Hence, part of the Cdk5 knockdown effect is mediated by Per2, and this is all what we can/want to say.

Additionally, our immunofluorescence experiments show, that knockdown of Cdk5 in the SCN prevents PER2 to be localized in the nucleus (Figure 6). If the reviewers view would be correct, this would mean that PER2 protein localization whether in the nucleus or not would be irrelevant for the generation of period length. This, however, is not what genetic experiments revealed, namely that PER2 is a negative feedback loop component of the circadian oscillatory mechanism important to generate circadian period (for review see Chong et al., 2012) for which its nuclear shuttling is essential.

However, the reviewer has a point, because this experiment shows that knockdown of Cdk5 can affect period not exclusively via PER2 and that there are other Cdk5 targets that may become important in absence of PER2 (what we actually state in the manuscript, see also the new Figure 7).

Per2^Brdm1^ still express a truncated form of Per2, hence the authors should also consider a similar experiment in Per2 KO animals to prove Per2-dependent mechanism.

Theoretically, it is correct that *Per2^Brdm1^* mice express a truncated form of PER2. This truncated form of PER2, however, is not in the nucleus, very unstable and degraded fast. We could not detect any PER2 protein in the SCN of *Per2^Brdm1^* mice (Figure 3) similar to our findings in Per2 total KO SCN (Chavan et al., 2016). This is the reason why the total Per2 ko and the *Per2^Brdm1^* mutant show the same free-running period (see Figure 2 and Chavan et al., 2016, Figure 2—figure supplement 5). Hence, a knockdown of Cdk5 in Per2 KO mice will show the same effect on period as the knockdown in *Per2^Brdm1^* animals.

Alternatively, the CDK5 phenotype in Per2^Brdm1^ animals could be due to CDK5 phosphorylating Per1, which is not discussed by the authors.

Since the PER1 and PER2 proteins only differ in a conservative amino acid change one amino acid before the PER2 Ser 394, i. e. PFD**Y**SPIR in Per2 and PFD**H**SPIR in PER1, it is likely that CDK5 may also phosphorylate PER1.

However, if PER1 would be phosphorylated at the mentioned site above at the same rate as PER2, then also PER1 in addition to PER2 would be absent in the nucleus. This would correspond to a PER1/PER2 double ko. PER1/PER2 ko mice become immediately arrhythmic when they are in constant darkness. This is not the phenotype we observe in the CDK5 knock-down mice and hence it is unlikely that Cdk5 has the same effect on PER1 as it has on PER2. The affinity and efficiency of CDK5 to phosphorylate Per2 may be much higher than that for Per1. However, if PER2 protein is absent (as in the Per2 Brdm mutant) PER1 may become phosphorylated by CDK5 to a greater extent (no competition with PER2) and affect period. Therefore, it will be of great interest to study the relationship between Cdk5, Per1 and Per2. We have discussed this aspect now in the manuscript.

- Figure 1. Eng, Edison and Virshup, 2017, demonstrated that CKIδ activity towards Per2 is regulated via phosphorylation of T347 by proline-directed kinases (presumably also CDK5). This possibility should be addressed by the authors.

We apologize that this study has escaped our attention. Although we mention that CDK5 may phosphorylate CKI (Ianes et al., 2016) this study also indicates this. However, both studies do not provide any evidence for CDK5 phosphorylating CKI at T347 in vivo. We cite now both articles in the revised version and depict this possibility in Figure 7.

- Figure 6. To prove the relevance of this mechanism to other peripheral oscillators, the authors should include an experiment in NIH-3T3 Bmal1-luc cells transfected with shRNA against CDK5.

This is a very good suggestion and we performed the proposed experiment in NIH-3T3 Bmal1-luc cells (see Author response image 1) previously in order to determine the efficiencies of CDK5 shRNAs. We observed a longer period, which would suggest that CDK5 activity in peripheral clocks in vitro has not the same effects compared to central clocks in vivo. CDK5 is a kinase that has many targets which are involved in many different signaling cascades. This may influence circadian period in central and peripheral clocks differently, although the molecular effect on PER2 as target is the same. For example, CDK5 phosphorylates CLOCK, which may be more important in peripheral tissues compared to the central pacemaker in the SCN, where Clock can be replaced by NPAS2. Since the results we present in the paper are showing the role of CDK5 in the central clock, we rather prefer not to put the result on peripheral clocks in vitro in this paper. This will avoid confusion between central and peripheral effects of CDK5 in the clock.

- Mutation of PER2 Ser 394 impacts circadian oscillation.

This is a very good suggestion, which we also wanted to do. However, overexpressing a PER Ser 394 mutant protein in cell cultures will simply interfere with the circadian clock mechanism in cells, because there is no system at hand that would allow us to express even the normal PER2 protein in a dynamic (circadian) manner. In other words, there is no cell culture system available that would rescue Per2 ko cell lines by overexpression of PER2, since the dynamic presence over time of PER2 is essential for establishing a circadian rhythm. Hence having no rescue system for wild type PER2, it makes no sense to overexpress Ser 394 mutant PER2. Its overexpression simply will interfere with the dynamic time dependent protein-protein interactions of the clock machinery and block it in a similar manner as wild type PER2 would (see Wallach et al., 2013, PLoS Genetics, 9, e1003398).

In vivo a knock-in experiment would be interesting to replace normal PER2 with a Ser 394 PER2 mutant protein. However, we would expect as similar phenotype as in the *Per2^Brdm1^* mutant since the Ser 394 is missing there as well and the PER protein in the *Per2^Brdm1^* mice is very unstable and degraded quickly. Furthermore, a PER2 Ser 394 knock in experiment in mice is for us not feasible to do within 2 months.

- CDK5 potentially also phosphorylates and regulates PER1 stability and transport.

This is a highly interesting question (see also above). Since the PER1 and PER2 proteins only differ in a conservative amino acid change one amino acid before the PER2 Ser 394, i.e. PFD**Y**SPIR in Per2 and PFD**H**SPIR in PER1, it is likely that CDK5 may also phosphorylate PER1.

However, if PER1 would be phosphorylated at the mentioned site above and its function would be the same as in PER2, then also PER1 in addition to PER2 would lack in the nucleus. This would correspond to a PER1/PER2 double ko. PER1/PER2 ko mice become immediately arrhythmic when they are in constant darkness. This is not the phenotype we observe in the CDK5 knock-down mice and hence it is unlikely that Cdk5, has the same functional effect on PER1 as it has on PER2. We agree that it remains to be seen what the function of a potential phosphorylation of PER1 by CDK5 might be. This is, however, out of the scope of this manuscript.

- CDK5 regulation of CLOCK (and potentially NPAS2) contributes to the observed phenotype: CLOCK stability and transport are impacted by Cdk5 knockout?

As the reviewer suggests, a study has indicated that CDK5 may phosphorylate CLOCK in vitro (which is cited in the text). If CDK5 regulation of CLOCK in the SCN would play a major role, we would expect the period in wheel running activity to become longer, similar to CLOCK mutant mice (Vitaterna et al., 1994, Science, 264: 719-25). What we observe, however, is a shorter circadian activity period upon knockdown of Cdk5. Therefore, it is unlikely that Clock plays a major role in the behavioral phenotype we observe. Whether NPAS2 can be phosphorylated by CDK5 is unknown and we cannot exclude such a possibility. Interestingly, NPAS2 does not show any consensus sequence that would be recognized by CDK5. Therefore, it is unlikely that CDK5 phosphorylates NPAS2. However, further investigation of the role of CDK5 in the phosphorylation of CLOCK may reveal additional and novel aspects in circadian clock regulation, but is out of the scope of this study.

Is the circadian effect of Cdk5 knockdown also observed in Clock knockout?

This is related to the question above and very interesting. We have not done an experiment in Clock knock-out mice with a knockdown of Cdk5. I would expect that the longer period phenotype of Clock KO mice would be shortened and appear normal like in wild-type. This is of course a speculation and it remains to be seen what happens in Clock KO mice. Knockdown of Cdk5 would also be interesting to perform in Npas2 KO, Cry KO, Per1 KO, Rev-erba KO and Bmal1 KO mouse strains. However, this is out of the scope of this study.

While the effect of CDK5 on circadian period is clearly showed, the involvement of PER2 in the mechanism is not clear, meaning the model on Figure 7 is mainly speculative at this stage.

From the questions and answers above it is clear that the regulation of the circadian clock mechanism by CDK5 is more complicated than depicted in the model. We acknowledge this in the revised Figure 7 highlighting the potential additional players that may be influenced by CDK5. However, some aspects in the model concerning PER2 are supported by our findings as well as by previous studies that are cited. A model is always based on what we know. Because we do not know everything of a process the model has to be continuously adapted according to new findings.